# Climatological moisture sources for the Western North American Monsoon through a Lagrangian approach: their influence on precipitation intensity

Paulina Ordoñez[1], Raquel Nieto[2], Luis Gimeno[2], Pedro Ribera[3], David Gallego[3], Carlos A. Ochoa[1], Arturo I. Quintanar[1]

[1]Centro de Ciencias de la Atmósfera, Universidad Nacional Autónoma de México, Mexico City, 04510, Mexico.
[2]Environmental Physics Laboratory (EphysLab), Universidade de Vigo, Ourense, 32004, Spain
[3]Dpto. de Sistemas Físicos, Químicos y Naturales, Universidad Pablo de Olavide, Seville, 41013, Spain.

*Correspondence to*: Paulina Ordoñez (orpep@atmosfera.unam.mx)

**Abstract.** This work examines the origin of atmospheric water vapor arriving to the Western North American Monsoon (WNAM) region over a 34-yr period (1981-2014) by using a Lagrangian approach. This methodology computes budgets of evaporation minus precipitation by calculating changes in the specific humidity of thousands of air particles advected into the study area by the observed winds. The length of the analyzed period (34-yr) permits the method to identify oceanic and terrestrial sources of moisture to the WNAM from a climatological perspective.

During the wet season, the WNAM region itself is on average the main evaporative source, followed by the Gulf of California. However, water vapor originating from the Caribbean Sea, the Gulf of Mexico and terrestrial eastern Mexico is found to influence regional-scale rainfall generation.

Enhanced (reduced) moisture transport from the Caribbean Sea and the Gulf of Mexico from 4 to 6 days before precipitation events seems to be responsible for increased (decreased) rainfall intensity on regional scales during the monsoon peak.

Westward propagating mid to upper level inverted troughs (IVs) seem to favor these water vapor fluxes from the east. In particular, a 200% increase in the moisture flux from the Caribbean Sea to the WNAM region is found to be followed a few days later by the occurrence of heavy precipitation in the WNAM area. Low-level troughs off the coast of northwestern Mexico and upper-level IVs over the Gulf of Mexico are related to these extreme rainfall days as well.

## 1 Introduction

Historical studies used the reversal in large-scale lower tropospheric circulation to identify a monsoon domain (Ramage, 1971). Such monsoon domains were found mainly over tropical areas of the Eastern Hemisphere because the seasonal wind
reversal is much better defined there than over the Americas (Hsu, 2016). Besides the wind field, precipitation is another fundamental variable that has been more recently used to define a monsoon climate, in which, due to the annual cycle of solar heating, the majority of the annual rainfall occurs in summer, while winters are quite dry. This rainfall-based classification of monsoon regions includes the North American and South American monsoon regions located roughly over the tropical to subtropical Americas. Particularly, the "North American monsoon (NAM)" region covers much of Central
America and central Mexico extending over northwestern Mexico almost to the southwestern United States (U.S.) (e.g. Wang and Ding, 2006; 2008; Liu et al., 2009; Wang et al., 2012; Huo-Po and Jia-Qi, 2013; Lee and Wang, 2014; Liu et al., 2016; Mohtadi et al., 2016; Wang et al., 2018).

The term NAM has also been extensively used to denote the monsoon-like summer rainfall of a more reduced region
centered over northwest Mexico and extending northward into the southwestern U.S. This NAM region covers approximately the north tip of the previous cited NAM region together with part of the adjacent Arizona and New Mexico (e.g., Douglas et al. 1993; Adams and Comrie 1997; Higgins et al. 1997; Barlow et al. 1998; Higgins et al. 1999; Higgins and Gochis, 2007). The onset of this NAM occurs over northwestern Mexico around mid to late June and over the southwestern U.S. around the beginning of July, at the same time as the development of a thermally induced trough,
northward displacements of the Pacific and Bermuda highs, and the formation of southerly low-level winds over the Gulf of California (GOC). Therefore, there is also a seasonal surface wind reversal, but it is not of sufficient magnitude and scale to meet the Ramage's criteria (Ramage, 1971; Hoell et al., 2016). This system is fully developed during July-August-early September (Vera et al., 2006) and supplies about 70%, 45% and 35% of the annual precipitation for northwestern Mexico, New Mexico, and Arizona, respectively (Vivoni et al., 2008; Erfani and Mitchell, 2014).

It is important to highlight that the same name is being employed in the scientific literature to denote the climatic characteristics of different regions. Henceforth in this work, the term "Western-North American Monsoon (WNAM)" is used to refer to the summer climate of northwestern Mexico and Southwestern U.S., distinguishing this monsoonal region with its own regional characters, from the larger NAM region which extends northward from the Equator.

The identification of the origin of the water available for precipitation in a region constitutes a very complex problem. Over the years, it was accepted that moist air moves into the WNAM system on a broad band of middle troposphere southeast

winds from the Gulf of Mexico (GOM) (Jurwitz, 1953; Bryson and Lowry, 1955; Green and Sellers, 1964). Later studies claimed that the eastern tropical Pacific and boundary layer flow from the GOC are the major sources of moisture for the WNAM system (Douglas, 1995; Stensrud et al. 1995; Berbery, 2001; Mitchell et al., 2002), while the middle tropospheric transport also remained important (Schmitz and Mullen, 1996). In addition to mean flow moisture transport, transient

features such as the "gulf surge", a coastally trapped disturbance that is typically initiated by a tropical easterly wave or tropical storm that crosses near the GOC entrance and is then propagated north-westward along the GOC axis (Rogers and Johnson, 2007; Newman and Johnson, 2013), are also important mechanisms for initiating precipitation in the WNAM. A gulf surge is termed wet or dry depending on if the surge is followed by positive or negative spatially averaged mean precipitation anomalies over Arizona and/or western New Mexico (Hales et al. 1972; Stensrud et al. 1997; Higgins et al.

2004; Pascale and Bordoni, 2016a). Wet surges occur between seven and ten times during the monsoon season (Pascale et al., 2016b). Transient upper-level inverted troughs (IVs), cold-core cut off lows, open troughs in the westerlies and surface fronts (Douglas and Engelhardt, 2007; Seastrand et al., 2015) also contribute to precipitation events in the WNAM. Gulf surges often occur in conjunction with such disturbances, particularly IVs, to produce rainfall over the northern WNAM region (Stensrud et al. 1997; Fuller and Stensrud, 2000; Higgins et al. 2004; Bieda et al. 2009; Newman and Johnson, 2012;

Seastrand et al. 2015).

Today it is widely accepted that both the middle level easterly moisture coming from the GOM and the southwesterly low-level moisture from the GOC contribute to monsoonal precipitation. In addition, other studies have highlighted the role of surface soil moisture and vegetation dynamics in the WNAM region (e.g., Dominguez et al., 2008; Mendez-Barroso et al.,

2009; Mendez-Barroso and Vivoni, 2010; Bohn and Vivoni, 2016; Xiang et al., 2018). Specifically, Hu and Dominguez (2015), using the extended dynamic recycling model (DRM, Dominguez et al., 2006) found that terrestrial sources contribute approximately 40% of monsoonal moisture. Bosilovich et al. (2003) used water vapor tracer diagnostics in global numerical simulations to quantify the effect of local continental evaporation on monsoon precipitation. These authors found that local evaporation is the second most important source of precipitation after the GOM. Dominguez et al. (2016) used water vapor

tracer diagnostics in a regional climate model to quantify the water vapor from four different oceanic and terrestrial regions that contribute to precipitation during the WNAM season. They have documented that local recycling is the second source after the lower-level moisture coming from the GOC. Therefore, despite the large number of studies of WNAM moisture sources, the major moisture sources to the WNAM system and their relative importance are still actively debated.

In this work, we use the Lagrangian particle dispersion model FLEXPART to analyze the water vapor transport towards the WNAM. Evaporation minus precipitation (E – P) is tracked from the WNAM region along the trajectories of appropriately selected particles, thereby facilitating the determination of water source-receptor relationships. This work addresses two main objectives: (1) define the main moisture sources for the WNAM region, and (2) determine the moisture transport that

contributes to the regional-scale rainfall intensity over the WNAM area. In Section 2 the data and methods are presented, followed by the results in Section 3. We dedicate Section 4 to the main conclusions of the study.

## 2. Data and Methods

### 2.1. Lagrangian diagnostic of E-P for the WNAM

This work makes use of the method developed by Stohl and James (2004; 2005) to quantify the atmospheric water vapor transport towards a region by using the Lagrangian particle dispersion model FLEXPART (Stohl et al., 2005) driven by meteorological gridded data. At the model start, the atmosphere is homogeneously divided into a large number of air parcels (particles), each representing a fraction of the total atmospheric mass. Then, the particles are allowed to move freely with the observed wind, overlapping stochastic turbulent and convective motions (Stohl et al., 2005) while maintaining their mass

constant. Particle positions and their specific humidity are recorded every 6 h. For each particle the net rate of change in water vapor content is computed by using the changes in specific humidity over time:

$$e - p = m\frac{dq}{dt},\tag{1}$$

where q is the specific humidity, m is the mass of the particle, and e and p are the rates of moisture increases and decreases of the particle along the trajectory, respectively. To diagnose the net surface water flux in an area A, the moisture changes of all particles in the atmospheric column over A are aggregated giving the field (E - P):

$$E - P = \frac{\sum_{k=1}^{K}(e-p)}{A},\tag{2}$$

where K is the number of particles residing over the area A, E is the evaporation rate and P is the precipitation rate per unit of area.

Finally, to find the moisture sources of a region, (e - p) of all the particles located in a given time over this region is

evaluated along their backtrajectories. By integrating the humidity changes (i.e., moisture increases and decreases) of all these particles, it is possible to find the areas where those particles have either gained (E – P>0) or lost moisture (E – P<0) along their path. It is also feasible to find the day of the recharging preceding the arrival at the target region. When a long enough period is analysed, the mean moisture sources can be described from a climatological point of view. Note that, as the particles originally located over the target region disperse, the particles residing in an atmospheric column do not represent

its entire atmospheric mass anymore, but only that part of the column fulfilling the criterion that it later reaches the target. (E

- P) values, therefore, do not represent the surface net water vapor flux, but only the net water vapor flux into the air mass travelling to the target region (Stohl and James, 2005).

In this study FLEXPART v9 was run for a 34-yr period from 1981 to 2014 driven by ERA-Interim reanalysis at 1ºx1º resolution (Dee et al., 2011) available on 61 model levels from 0.1 to 1000 hPa. There are approximately 14 model levels below 1500 m and 23 below 5000 m. This 34-yr period was the available period of data at the moment that the experiment was executed. We used analyses every 6 h (0000, 0600, 1200, and 1800 UTC) and 3-h forecasts at intermediate times (0300, 0900, 1500, and 2100 UTC). The 3-h forecasts are used here to supplement the analyses because the time resolution is critical for the accuracy of Lagrangian trajectories (Stohl et al., 1995). The above described method was applied backward in time from the WNAM region shown in the Figure 1. The boundary selected to represent the WNAM region is similar to that used by Hu and Dominguez (2015), whose core WNAM region was also consistent with the North American Monsoon Experiment (NAME: Higgins and Gochis, 2007) and NAME precipitation zones defined by Castro et al. (2012).

To find out the transport time adequately represented by FLEXPART, we used the CHIRPS database (Funk et al., 2015). CHIRPS integrates 0.05° resolution satellite imagery with in-situ station data and it was shown by Perdigon-Morales (2018) to properly reproduce some of the particular characteristics of the Mexican rainfall such as the mid-summer drought. The minimum absolute difference between the precipitation simulated by FLEXPART and the "real" precipitation over the WNAM defined by CHIRPS was 6 days, so we limited the transport time to this period. This lifespan was also computed with ERA-Interim precipitation data with similar results. Further details about this methodology can be found in Miralles et al. (2016).

### 2.1.1. Limitations of FLEXPART

FLEXPART requires only self-consistent meteorological analysis data as input. The accuracy of the employed data is critical, as errors in these data can lead to systematic miscalculations of (E - P). For instance, as the flux (E - P) is diagnosed using the time derivate of humidity, unrealistic fluctuations of humidity could be identified as water vapor fluxes. If these fluctuations are random, they will cancel over longer periods of time. However, if the trajectory data suffer from substantial inaccuracies, even if these errors are random, results can be systematically affected. This could be the case if, for instance, a particle that is originally located in a relatively moist air mass leaves this air mass due to trajectory errors and enters into a drier air mass. The humidity would then decrease along this trajectory and (E - P) < 0 would be erroneously diagnosed. The opposite is true for relatively dry air masses, i.e., (E - P) would be systematically too large for the tracking from dry regions (Stohl and James, 2005). In this sense, the ERA-Interim reanalysis used in this study has been found to provide a reliable representation of the atmospheric branch of the hydrological cycle when compared to other reanalysis products such as CFSR or MERRA (Trenberth et al. 2011; Lorenz and Kunstmann, 2012).

A second limitation is imposed by computational constraints. We executed FLEXPART driven by ERA-Interim data at 1°x1° spatial resolution. Global-scale atmospheric models with this grid spacing do not resolve convective clouds, even the mesoscale convective systems with horizontal dimensions on the order of a couple of hundred kilometres are not sufficiently

resolved and they must be parameterized (Foster et al., 2007). However, analogous works on moisture transport diagnosis have achieved promising results for tropical regions where convective rainfall clearly dominates, using this relatively coarse resolution (e.g., Duran-Quesada 2010; Drumond et al., 2011; Hoyos et al., 2018). Notwithstanding, given this limitation, we opted for not evaluating processes that occur over the WNAM at the local scale (sub-grid) or over a very short time (sub-daily).

Traditionally, the search for the origin of precipitation has been approached by the so-called Eulerian methods, based on the analysis of the divergent part of the vertically integrated moisture flux (VIMF). Eulerian methods are considered quite accurate at approximating E – P (Simmonds et al., 1999; Ruprecht and Kahl, 2003; Mo et al., 2005), however for this study we opted for using a Lagrangian approach for two main reasons. First, Stohl and James (2005) obtained practically identical

results using a Lagrangian approach based on FLEXPART and the Eulerian equivalent. Second, a Lagrangian approach allows forward or backward tracing along defined trajectories, facilitating the determination of the source–receptor water vapor relationships. Notwithstanding, in order to asses our results, here we make use of the VIMF to validate the moisture budgets calculated using FLEXPART. We have based this comparison both for ERA-Interim data and with the more recent "Modern-Era Retrospective analysis for Research and Applications V2" dataset (MERRA-2. Bosilovich et al., 2017).

MERRA-2 has been chosen because it incorporates an improved water cycle.

## 2.2. Tracking E-P for individual precipitation events

The methodology described in Section 2.1 determines the average net changes of q for air particles aimed to the study area, but the moisture transported towards the WNAM region does not always generate effective precipitation. This limitation can

be overcome by tracking the air particles that arrive to the WNAM during wet and dry days separately. Therefore, a definition of wet and dry days over the WNAM region is necessary. For this purpose, daily CHIRPS data are used with a spatial resolution of 0.25° x 0.25° and the assessment of long-term statistics is also performed on a 34-year (1981-2014) period. A "common wet day" for the WNAM is defined as a rainfall day covering a large proportion of the study area. This is achieved by using the methodology described in Ordoñez et al. (2012). At each grid point, precipitation values above 10%

of the standard deviation computed for all the grid points of the study area are considered individual precipitation events for that day. Next, the percentage of precipitation days for each grid point is computed and the averaged value is then obtained for all grids (20.3%). To define a wet day across the WNAM region we compute the percentage of grid points inside the region that must have simultaneous precipitation in order to obtain the annual value of average precipitation over the region.

This percentage resulted in 41.3% and a total of 2513 days were classified as wet days for the entire WNAM during the study period. Figure 2 shows the precipitation distribution throughout the year according to this methodology.

An analogous method is employed to classify the wet days according to their intensity. To define moderate and extreme precipitation events, the 50th (P50) and 90th (P90) percentiles of the precipitation time series at each grid point have been computed. Then, we require that these percentiles must be exceeded simultaneously in at least 41.3% of the grid points inside the WNAM area. In this way, the method assures that a moderate or extreme rainfall event is characterized by a well-determined precipitation value covering a significant portion of the WNAM region. Weak precipitation days are the remaining wet days. Figure 3 shows the yearly precipitation composites for the different precipitation categories. The general increase in the precipitation area as precipitation intensity increases for wet days is clearly appreciated in Figures 3b to 3d.

The accuracy of FLEXPART to capture rainy days in relation to CHIRPS is also tested by comparing the average (E - P) values obtained by FLEXPART over the WNAM domain during first 6-hr time step of the trajectories for the different precipitation events with the average values obtained using CHIRPS. For making this comparison we are assuming that E and P cannot coexist in the same point of space and time. Under this assumption, the instantaneous rates of evaporation or precipitation can be diagnosed by FLEXPART. Note that FLEXPART fails to diagnose precipitation during the weak and moderate rainfall events, yielding low positive (E - P) values. This result indicates that not all the moisture particles traced by FLEXPART are contributing to the precipitation events. However, the annual mean of (E - P) in the WNAM diagnosed by FLEXPART during extreme precipitation events is below -11.4 mm day$^{-1}$, where CHIRPS indicates values below 9 mm day$^{-1}$ of rainfall in most of the grid cells (see Figure 3d). This suggests that FLEXPART is reliable for capturing extreme precipitation events over the region.

In order to estimate the actual evaporation over the moisture source regions we used the state-of-the-art Global Land Evaporation Amsterdam Model (GLEAM). The monthly evaporation from the land was estimated from GLEAM v3.2 data at 0.25ºx0.25º which is largely driven by satellite data (Miralles et al., 2011).

We additionally use different climatic fields (geopotential height, specific humidity and horizontal winds components) from the ERA- Interim reanalysis in order to extract information about regional-scale patterns associated with the different rainfall intensity categories over the WNAM region defined above.

## 3. Results

### 3.1. Moisture sources for the WNAM region

Figure 4 shows the 6-day aggregated monthly average values of water vapor flux (E - P) before air masses aimed towards the WNAM reach the region for the period 1981-2014. The $(E – P)_n$ designates the water vapor flux value for day "n" before arrival to the target area. The sum of the net water vapor flux from day 1 to day 6 (sum of $(E – P)_1$, $(E – P)_2$,…, $(E – P)_6$) is denoted using $(E – P)_{1–6}$. Although the WNAM season is usually defined from July to September, we found several regional-scale precipitation events during June (see Figure 2), so the results for the 4 months from June to September are presented in this figure. Reddish (bluish) colors are used to show regions of water vapor gain, E – P>0 (loss, E – P<0), according to the sign of dq/dt of particles following their trajectories. The North Eastern Pacific off the coast of the United States and the GOC are found to be net moisture sources during the summer. The monthly analysis shows that the terrestrial region east of the WNAM area is also an active source throughout the summer. In addition, there are source regions over the GOM and the Caribbean Sea that seem to be significant mainly during July and August. The southwestern U.S. also appears as a source region from June to September, where evaporation is larger than precipitation. Finally, the WNAM region itself seems to be an evaporative moisture source for the whole region in June and September, whereas during July and August only the northern WNAM acts as an evaporative moisture source, while the southern WNAM indicates negative values of $(E – P)_{1–6}$, suggesting that this area is a sink of moisture during the peak monsoon.

Figure 5 depicts the average VIMF divergence for the same months (June to September) for the study period using ERA-Interim reanalysis at 1°x1°. Positive values indicate moisture flux divergence (E – P>0) while negative values indicate moisture flux convergence (E – P<0). The Eulerian results are quite similar to the Lagrangian diagnostics. Even the temporal variability over the eastern WNAM, the GOM and the Caribbean Sea is very similar, showing higher contributions during July and August compared to June and September. The main difference is seen over the Pacific Ocean, where the Eulerian method indicates VIMF divergence that does not appear as a moisture source on the Lagrangian approach. We found, however, that the moisture flux over this oceanic region is not aimed toward the WNAM. Therefore, the agreement between the Lagrangian and the Eulerian diagnostics is excellent. The Eulerian diagnostic performed with MERRA-2 data at 1.25° x 1.25° (not shown) does not capture the seasonal variability over eastern Mexico and the Caribbean Sea, but otherwise the monthly VIMF divergence patterns are consistent with those from ERA-Interim.

According to these results, six main moisture sources for the WNAM region have been defined: (1) the WNAM region itself (WNAM), (2) the terrestrial region east of the WNAM (NE-MEX), (3) the Atlantic which includes a part of the GOM and the Caribbean Sea (GOM-CAR), (4) the southwestern U.S., toward the north of the WNAM (SW-US), and the Pacific which includes (5) the Northeast Pacific (NEP) and (6) the GOC (GOC) . Figure 6 shows their boundaries. These source regions were defined using the values greater than P90 of $(E - P)_{1–6}>0$ for the period from June to September. The summer monthly

evolution of $(E - P)_{1-6}$ integrated for these fixed areas is shown in Figure 7. During June, the inland evaporative source over the WNAM itself and the water vapor from GOC are the main moisture sources for the WNAM System. In July, SW-US provides a slightly greater amount of moisture than these regions. The situation is a little different for August, when the WNAM region is the main source and the NE-MEX region shows its peak values of $(E - P)_{1-6}$. September is characterized by the peak contribution from the WNAM region, while the relative contribution from the remaining sources decrease with respect to their August values.

Following the monsoon onset (from July to September), FLEXPART describes a terrestrial moisture contribution from the WNAM region of 38% on average, with a water vapor flux from this region of 26%, 34% and 55% during July, August and September, respectively. Our results using FLEXPART find larger moisture transports from the WNAM region than other previous studies, but our results are not strictly comparable to them. For instance, Bosilovich et al. (2003) considered the entire Mexican continental region and found it to be the dominant source of moisture to the monsoon, with contributions of roughly 30%, 25% and 20% during July, August and September, respectively. In their study these authors computed the fraction of precipitation that originates as evaporation and estimated the values of evaporation from all the Mexican territory that contributes to the WNAM precipitation. Hu and Dominguez (2015) estimated the precipitable water contribution from recycling to be about 10% during the monsoon peak. But the model they used for this work has proven to be imprecise at tracking the moisture transport in the monsoon region because of the model's assumption of a well-mixed atmosphere. This assumption does not holdover the WNAM region where a relatively strong shear occurs, causing an underestimation of local recycling by their model (Dominguez et al., 2016). As previously stated, FLEXPART estimates the net moisture gain, $(E - P) > 0$, since precipitation and evaporation are not directly separable in this model.

Regarding the summer monsoon evolution of this inland evaporative source, Bosilovich et al. (2003) affirm that the terrestrial supply of moisture to the WNAM decreases with time, while Hu and Dominguez (2015) show a maximum local contribution during August. In our case, (E - P) considerably increases with time throughout the summer monsoon (Figure 5). GLEAM shows that the greatest monthly mean evaporation over the WNAM during the summer monsoon season occurs in August followed by July and September. The observed monthly mean precipitation peaks in July decreasing with time. The (E - P) monthly means from the difference of these independently estimated values of E and P is lowest in July and increases with time peaking in September, consistently with our results.

### 3.2. Role of moisture source regions during regional scale precipitation events

Figure 8 depicts variability in the advected moisture from the different source regions between wet and dry days during the monsoon season (JAS). The time series of (E - P) for the source regions in Figure 6 are shown from the sixth to the first day before the air particles aimed at the WNAM reach this region.

The WNAM region contributes with higher moisture amounts before the rainfall days, except for the day -1 when a fraction of the particles located over the region should be already losing moisture. Similar conditions are experienced by the adjacent GOC and NE-MEX regions, both regions supply moisture before day 0, but due to the proximity to the target region, the air masses start to lose part of their moisture the day before the arrival (day -1). In the case of NE-MEX, $(E - P)_{-1} < 0$ which means that this region is a net sink of moisture, what make sense as the air masses have to cross the mountain range Sierra Madre Occidental into the WNAM region. GOM-CAR provides higher moisture amounts from day -6 to -3, while during days -3 to -1 the particles still do not reach this region. SW-US also could contribute to rainfall development as this region as supplies more water vapor to the WNAM before day 0. Finally, the NEP region shows slightly lower moisture contributions before day 0, indicating that the moisture contribution from the GOC is coming throughout the south.

Although all the regions excluding the NEP could potentially contribute to the rainfall generation over the WNAM, the greatest difference in the total amount of moisture transported before dry and wet days is obtained for the NE-MEX area. Bosilovich et al. (2003) found the dominant sources of monsoon precipitation to be the local evaporation and transport from the tropical Atlantic Ocean (including the GOM and Caribbean Sea). In contrast, Dominguez et al. (2016) reported that the GOC contributes with a higher moisture amount to the WNAM precipitation than the GOM and the local evapotranspiration (ET). This latter study used water vapor tracers embedded into a regional climate model using lateral boundary conditions derived from the North American Regional Reanalysis (NARR; Mesinger et al. 2006). However, the ET fields from NARR might have significant deficiencies in the primary moisture source areas (Bohn and Vivoni, 2016). Our results are in better agreement with those of Bosilovich et al (2003) and suggest that the GOM-CAR and NE-MEX could be a major contributor to the monsoonal rainfall. However, we cannot conclude that this is one decisive source for rainfall development, because as we have mentioned, the other sources also exhibit important changes for precipitation versus no precipitation days.

Our next objective is to examine which source regions are the most relevant for the modulation of rainfall intensity. We have found only 32 extreme precipitation events occurring in September from 1981 to 2014. Therefore, in order to obtain statistically meaningful results, only water vapor transport that generated heavy rainfall during the monsoon peak (July and August) is studied. Figure 9 shows the difference of (E - P) for the extreme precipitation days with respect to weak intensity days for days -1, -2 and -5. For the northern WNAM, $(E - P)_{-1}$ is lower just before the extreme days than before weak days, indicating greater moisture loss preceding extreme days. On day -2, larger evaporation is seen before the extreme days over the WNAM region as a whole. Five days preceding an extreme or weak day, larger values of $(E - P)_{-5}$ are found along a pathway that crosses the NE-MEX and reaches the GOM-CAR region for extreme days relative to weak days. A high (low) moisture supply from the GOM-CAR for 4 to 6 days back in time seems to be one of the most important factors affecting the precipitation intensity, particularly over the northern part of the WNAM (see Figure 3) during the monsoon peak.

The above-mentioned differences can be clearly observed in Figure 10, which shows the integration of the moisture changes for the WNAM, NE-MEX and GOM-CAR before the extreme, moderate and weak rainfall days. The integration over the GOC is also depicted as it is a relevant moisture source for the WNAM. Over GOM-CAR and NE-MEX, the water intake from day -6 to -4 and from day -6 to -2 respectively, seems to be related to the precipitation intensity over the WNAM (Figures 10c and 10b), with an increase of more than 200% in the case of GOM-CAR and 44% for NE-MEX from weak to extreme rainfall days. The integrated time series of (E - P) for WNAM itself show that $(E - P)_{-1}$ and $(E - P)_{-2}$ for weak precipitation days (Figure 10a) present a behavior similar to dry days (Figure 8a). In contrast, moderate and heavy rainfall events show a completely different behavior that could be related to the strong surface heating that is needed prior to such precipitation events. In the case of the GOC, it is noteworthy that the changes in the moisture transport amount are inversely related to the rainfall intensity at day -1 (Figure 10d), which could be associated with the proximity with the target region. These results suggest the southeasterly vapor fluxes from the Caribbean Sea, overtaking the Sierra Madre at higher latitudes, are related to the monsoonal rainfall intensity.

Figure 11 depicts geopotential height and moisture transport differences at 700 and 200 hPa, for weak, moderate and strong precipitation days with respect to dry days. For the low to mid-troposphere (Figures 11a, 11b, and 11c) positive geopotential height differences show a center over the western U.S. and transient lows (easterly waves or tropical cyclones) off the west coast of Mexico are also observed. Both features are better defined as the precipitation becomes more intense over the WNAM. At 200 hPa, a positive height difference is located over the desert Southwest as an extension of the monsoon anticyclone (Figures 11d, 11e and 11f). A negative height difference develops roughly over the GOM that is also more intense as the precipitation intensifies over the WNAM and may indicate an inverted trough (IV). This result has been previously described as low-level troughs interacting with an upper-level IV enhancing precipitation into the southwestern United States and northwest Mexico (Stensrud et al. 1997; Fuller and Stensrud 2000; Higgins et al. 2004; Seastrand et al., 2015). IVs have been associated with heavy rainfall events in the border region of U.S. and Mexico. (Bieda et al. 2009; Finch and Johnson, 2010; Newman and Johnson 2012). How exactly mesoscale and synoptic circulations related to IVs help organize deep convection over the NAM region is not entirely known (Lahmers et al., 2016). Newman and Johnson (2012) find that these transient features increase surface-to-mid level wind shear, with midlevel flow from the northeast perpendicular to the topography. The enhanced vertical wind shear across the topography supports the upscale growth and westward propagation of diurnal convection initiated over the Sierra Madre Occidental, resulting in wide spread convection over the western slopes and coastal low lands of the WNAM region. Divergence aloft on the west flank of an IV can also lead to ascent and destabilization (Pytlak et al., 2005). Regardless of the physical mechanisms, these composites support previous studies that IVs play an important role in generating widespread heavy precipitation across the WNAM region.

More observations of the dynamic and thermodynamic environment during the passage of IVs, as well as improved models of the flow over complex terrain are both needed to better understand the role of IVs in supporting convective outbreaks

across the monsoon region. This work suggests an anomalous tongue of midlevel moisture over the northeastern Mexico (Figure 9c) occurs in conjunction with upper-level IVs and is related to wide spread heavy precipitation over the WNAM region.

## 4. Discussion and conclusions

Despite the large body of literature on the transport of water vapor and precipitation patterns associated with the WNAM, there are still important knowledge gaps regarding the sources of water vapor, their relative importance, and the detailed pathways through which the water vapor can reach the WNAM region. This study is focused on the large-scale aspects of moisture transport and precipitation occurrence over the WNAM. The well-tested FLEXPART model is used to assess the location of the major moisture sources of the WNAM for a 34-yr period from 1981 to 2014. Six main moisture sources have been identified, three terrestrial and other three oceanic: the evapotranspiration from the region itself, the Mexican terrestrial area east of the WNAM region, the southern part of the Gulf of Mexico and the adjacent Caribbean Sea, the southwestern U.S., the Gulf of California and the most eastern part of the Northern Pacific. The main moisture sources identified by FLEXPART coincide with the existent literature, in which traditionally the debate has been centered on the relative importance of the Gulf of California versus the Gulf of Mexico and more recently, on the role of the recycling process.

Our results indicate that during the monsoon season (from July to September), the WNAM itself is the main moisture source, while the Gulf of California is the second one in importance. However, when the moisture transport for the days leading up to regional-scale wet and dry days are compared, the relevance of the water vapor originated at the Caribbean Sea and the Gulf of Mexico on days -4 to -6 is clearly evidenced. A clear difference in E - P is seen between extreme and low precipitation days over the Gulf of Mexico, the Caribbean Sea and the terrestrial area east of the WNAM region prior to the onset of precipitation, suggesting that these regions could play a significant role in supporting regional-scale heavy precipitation development over the WNAM. On the other hand, the relevance of the water vapor transport from the Gulf of California diminishes a day before regional-scale precipitation events, as the intensity of both the low level transient lows and the precipitation over the WNAM increase.

It must be stressed that currently there is a lively debate relative to the origin of the water vapor advected to northern WNAM leading to extreme precipitation events. In a recent study, Ralph and Galarneau (2017) documented the role of the transport of water vapor from the east in modulating the most extreme precipitation events over southeastern Arizona. The water vapor aimed to this region was hypothesized to flow through a gap in the mountain range that connects the Continental Divide and the Sierra Madre in southern Arizona-New Mexico and Northern Mexico, named the Chiricahua Gap. In contrast, Jana et al. (2018) found the Gulf of California as the leading moisture source for precipitation development at two locations

selected over Arizona (Laveen) and New Mexico (Redrock), although these authors also find a moisture contribution from the Gulf of Mexico at low levels (below 2000 m) for a western New Mexico location.

Due to the limitations of our methodology, our results are not conclusive, but they seem to support the significance of the westward moisture flux from the Gulf of Mexico and the Caribbean Sea in the extreme WNAM precipitation. We cannot
5   assure that this anomalous water vapor transport implies air masses crossing Sierra Madre into the WNAM region developing strong precipitation, as this process cannot be resolved at the spatial resolution use and moreover, low-level moisture is also needed to develop convection. However, we have found the presence of low-level troughs and upper-level IVs in these same extreme wet days, giving a scenario compatible with convective precipitation occurrence. In this sense, Schiffer and Nesbitt (2012) describe deep easterly flow anomalies along the southern edge of the monsoon high over the
10   WNAM core prior to an initiation of a wet surge that would be essential for providing pre-surge moisture to the northern WNAM. Whereas local recycling and GOC moisture would be important after the surge arrives at the northern end of the GOC. Other authors also reported gulf surges occurring if an easterly wave trough passes east to the GOC following the passage of an upper-level mid-latitude trough (Stensrud et al., 1997; Fuller and Stensrud, 2000). Our results agree with these works, suggesting that moisture from the GOM could be important together with other sources such as tropical cyclones and
15   IVs, which sometimes can even form from midlatitude fronts before propagating westward in the easterly flow south of the upper tropospheric monsoon anticyclone (Lahmers et al., 2016). These features would imply that the WNAM cloud be considered as a hybrid monsoon, with characteristics of a tropical monsoon, but also with impacts from mid-latitudes.

**Competing interests**

The authors declare that they have no conflict of interest.

**Acknowledgments**

This research is a contribution to the project PAPIIT IA103116 "Principales fuentes de humedad de la República Mexicana y

su variabilidad climática". Pedro Ribera and David Gallego were also supported by the Spanish "Ministerio de Economia y Competitividad" through project CGL2016-78562-P and by research group RNM-356 belonging to the "Plan Andaluz de Investigación Desarrollo e Innovación". We thank Yolande Serra for in depth discussions and her comments on the manuscript.

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

**Figures**

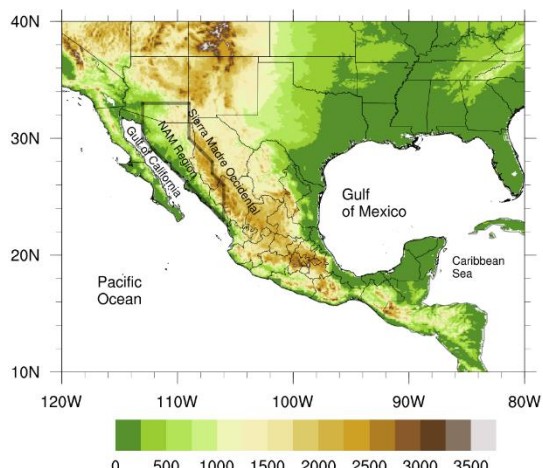

**Figure 1: Study region (black solid line) and its topography (m).**

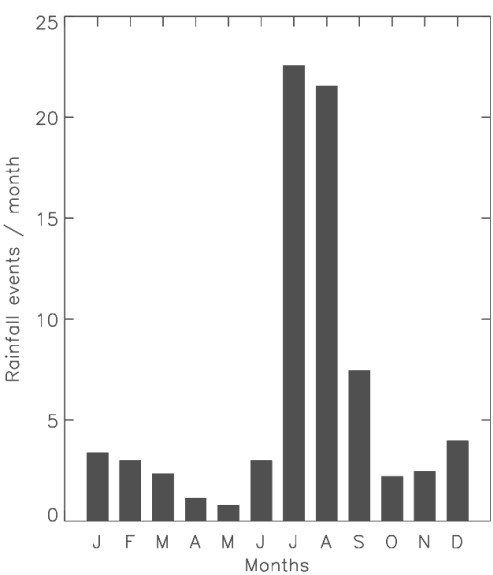

**Figure 2. Number of precipitation events per month.**

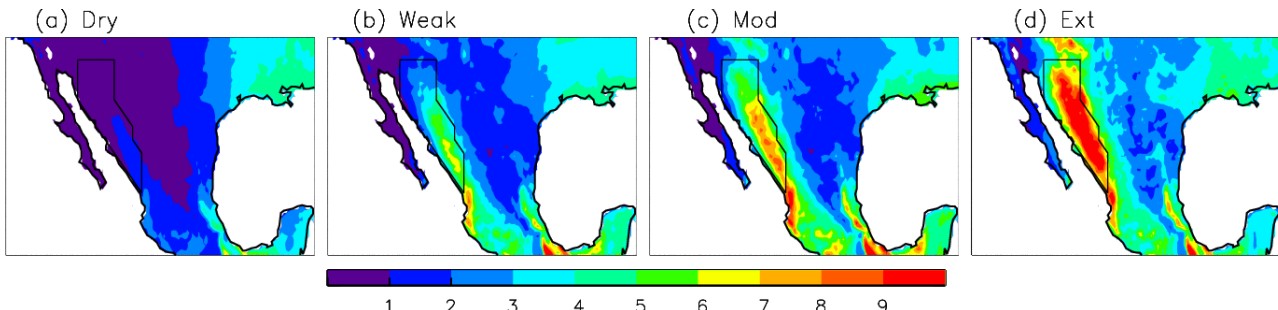

**Figure 3. Mean daily rainfall (mm/day) (1981-2014) for the (a) dry days, (b) weak precipitation days, (c) moderate precipitation days, and (d) extreme precipitation days over the WNAM. Black boundary delineates the study region.**

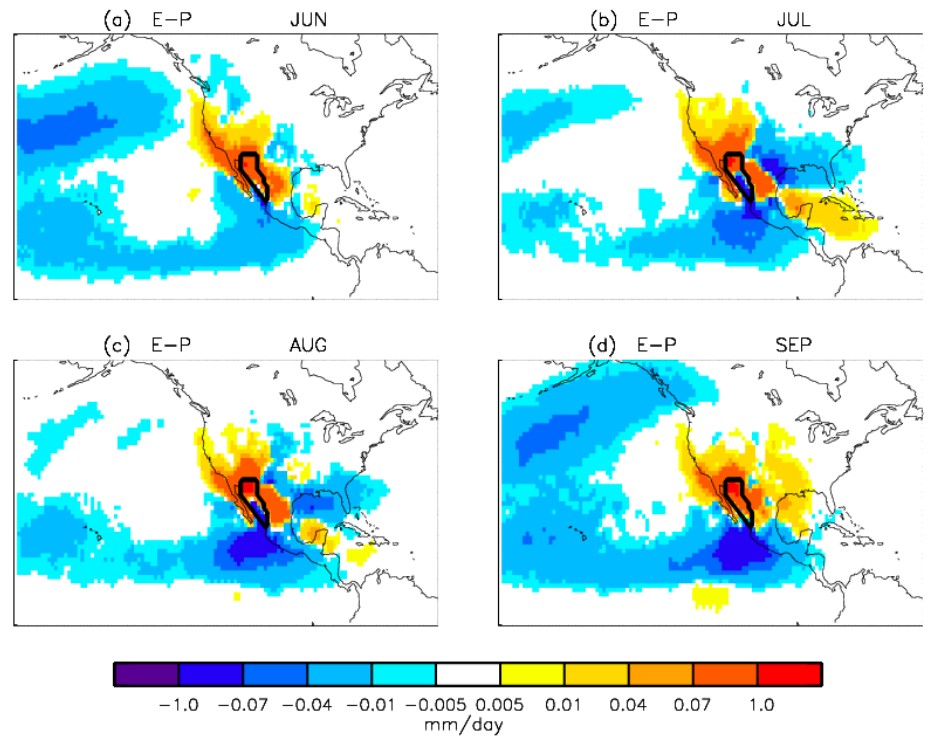

**Figure 4. Monthly averaged values of (E – P)$_{1-6}$ (mm/day) for all the particles aimed toward the WNAM region during (a) June, (b) July (c) August and (d) September (period of study: 1981 – 2014). Black boundary delineates the study region.**

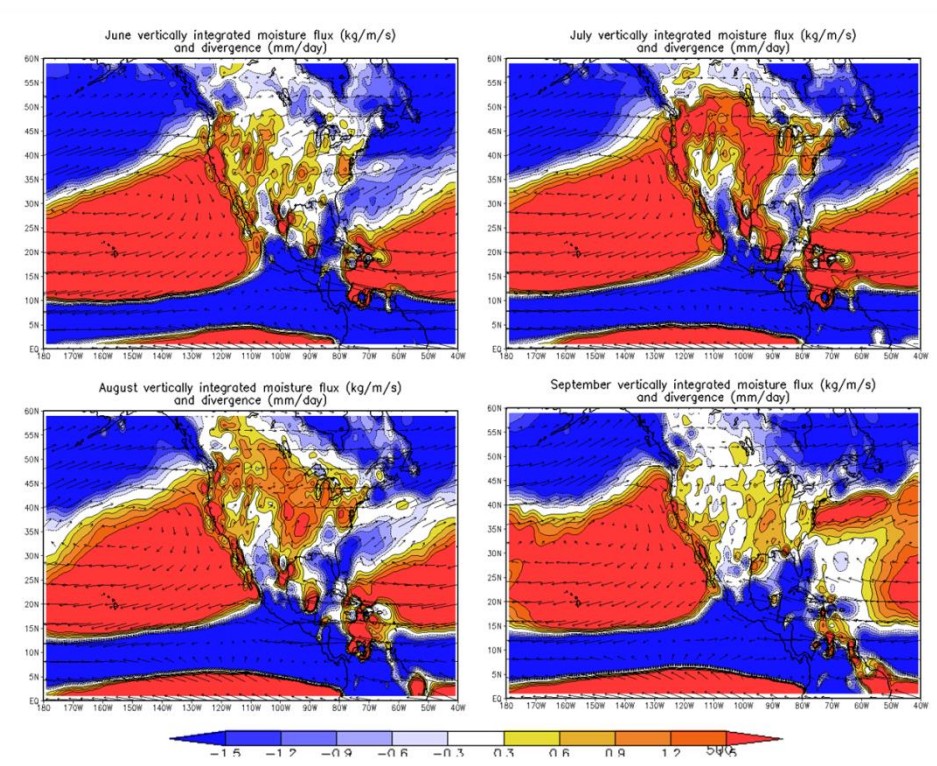

**Figure 5. Monthly averaged values of vertically integrated moisture flux (kg m⁻¹ s⁻¹) and divergence–convergence (reddish-bluish colors) (mm day⁻¹).**

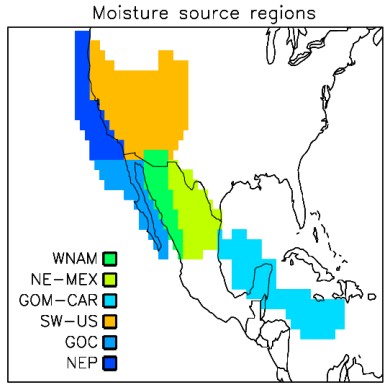

**Figure 6. Name and geographic limits of the moisture sources defined for the WNAM region.**

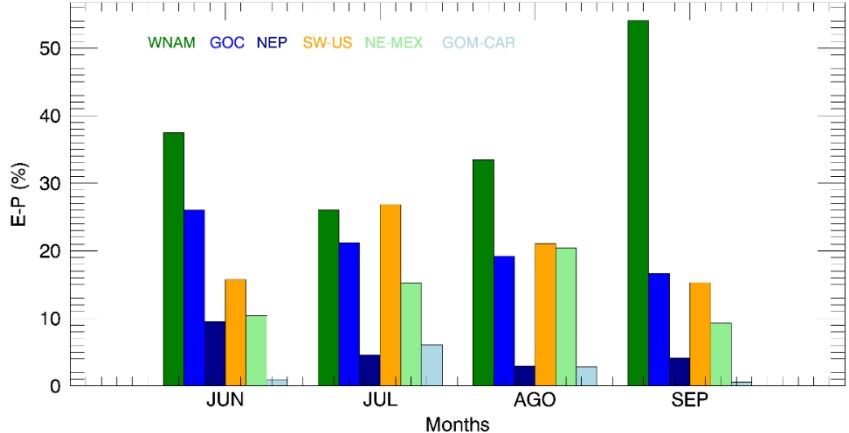

5 **Figure 7. Monthly (E − P)$_{1-6}$ percentages for the six areas defined as moisture sources.**

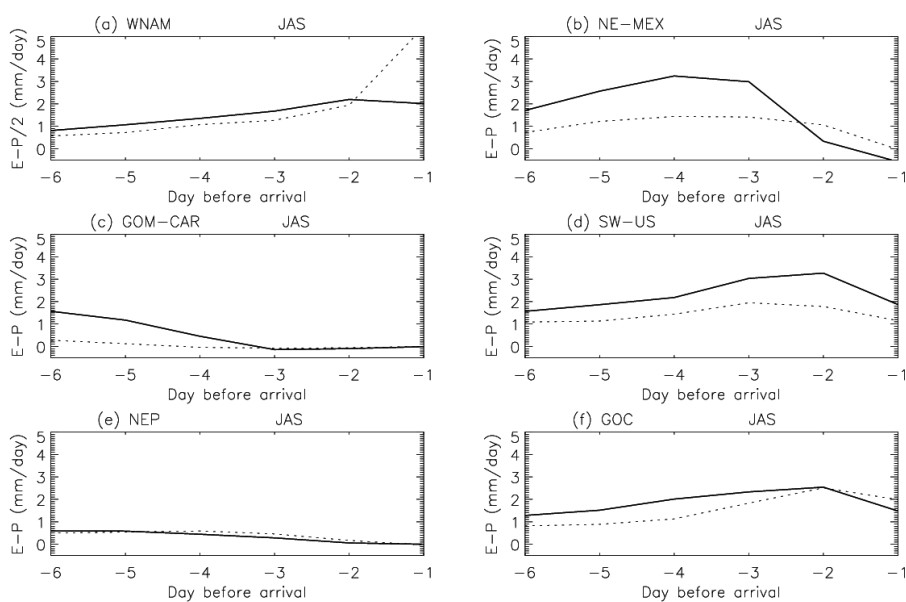

10 **Figure 8. JAS time series of (E − P)$_n$ (n=1 to 6) integrated over (a) WNAM, (b) NE-MEX, (c) GOM-CAR, (d) SW-US, (e) NEP and (f) GOC. Solid line: wet days. Dotted line: dry days. Note that the panel a is scaled by 0.5.**

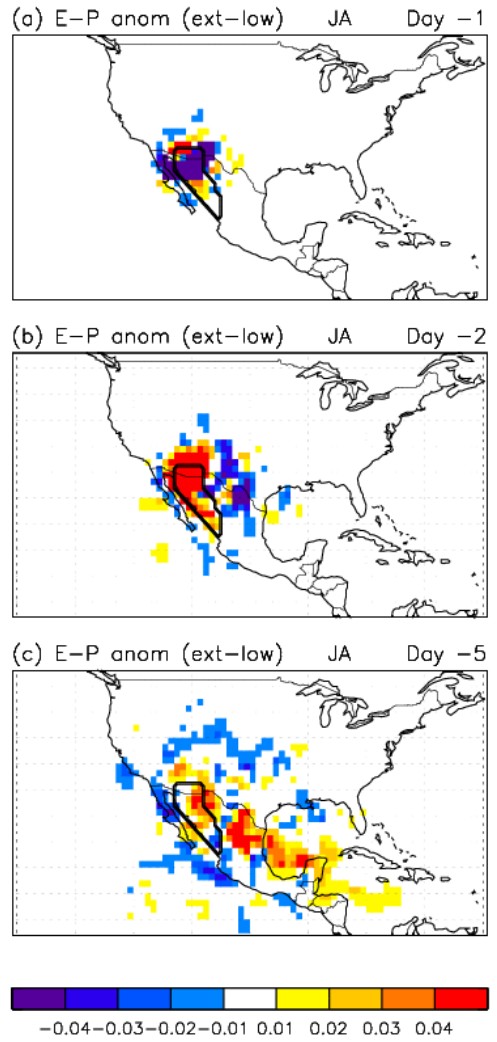

**Figure 9. Anomalies of $(E - P)_1$, $(E - P)_2$, and $(E - P)_{-5}$ during JA (1981–2014) for extreme rainfall days minus low rainfall days. Unit: mm/day. Black line delineates the study region.**

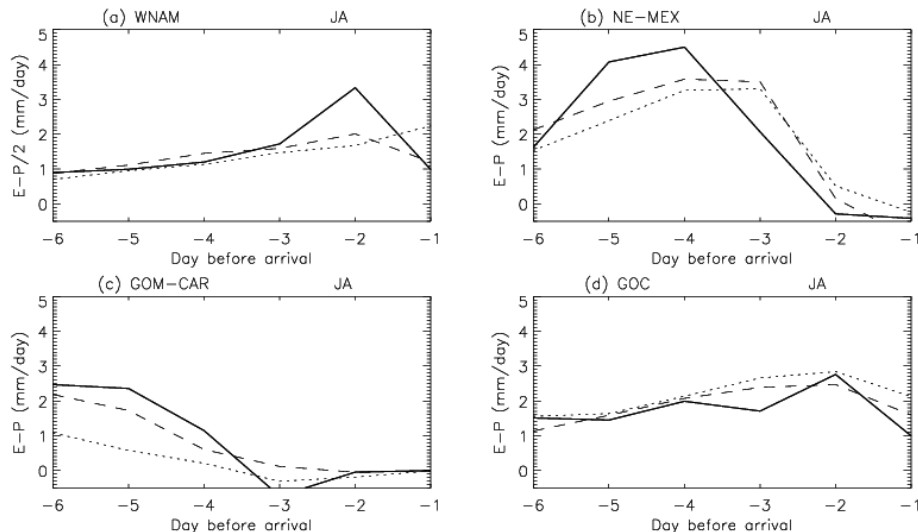

**Figure 10.** JA time series of $(E-P)_n$ (n=1 to 6) integrated over (a) WNAM, (b) NE-MEX, (c) GOM-CAR and (d) GOC. Black solid line: extreme, dashed line: moderate and dotted line: weak rainfall events. Note that the panel a is scaled by 0.5.

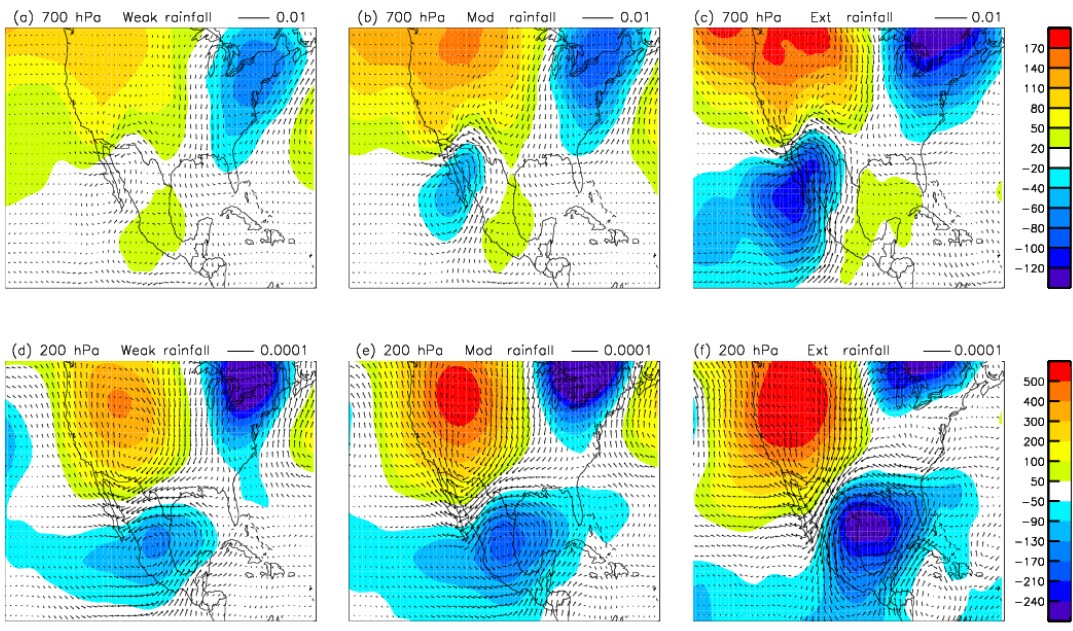

**Figure 11.** Composites at of geopotential height anomalies (colors, m) and moisture transport anomalies (arrows, kg kg$^{-1}$m s$^{-1}$) with respect to regional-scale dry events over the WNAM during the monsoon peak (July and August) for (a) weak, (b) moderate and (c) extreme precipitation events at 200 hPa and (d) weak, (e) moderate and (f) extreme precipitation events at 700 hPa.