# Peer review of "Climatological moisture sources for the Western North American Monsoon through a Lagrangian approach: their influence on precipitation intensity"

_Earth System Dynamics, 2018_

## Referee Comment (RC1) · E. R. Vivoni (Referee) · 2 Jun 2018

General Comments:

The study describes the sources of atmospheric water vapor during the North American monsoon and its dependence on the size of synoptic-scale rainfall periods. To achieve this, a particle tracking analysis of air parcels was applied using a reanalysis product in order to estimate fields of evapotranspiration minus precipitation (E-P) taken as the metric to study sources and sinks of water vapor. The authors have done a very nice job in bringing a different tool to a subject matter that is under considerable debate. Their work provides some new insights that will be useful for the community at

large to consider. Furthermore, their work is well written and illustrated. The comments below are intended for the authors to improve their work and increase its overall impact by demonstrating the solid nature of their method, results and interpretations.

Specific Comments:

1. It is important that the authors have a more in-depth explanation of the FLEX-PART algorithm as it pertains to the generation of the E-P field, including the limitations therein, within the introduction and methods section. Some of the limitations are indicated subsequently in the results section, which is considered too late in the manuscript. What type of errors in E-P are expected from its estimation as dq/dt? How sensitive is the method to the selection of the ERA-Interim reanalysis fields at 1 degree, 3 hourly resolution? How realistic are these fields with respect to observations or other reanalysis products that independently estimate P and E, and from which E-P can be obtained? The reader needs to have confidence in the accuracy of E-P before it is used to make inferences on the sources and sinks of water vapor in the monsoon region.

2. It would be useful for the authors to present a justification and/or further detailed explanation for the following aspects of their methodology:

a. The coarse resolution (1 degree by 1 degree) of the meteorological fields used in the FLEXPART model, given the scale of land and ocean features in the NAM region.

b. The selection of the time period (1981-2014).

c. The boundary selected to represent the NAM region and its consistency with the tiered approach advocated by Higgins and co-authors during the North American Monsoon Experiment.

d. The use of the term anomaly when discussing differences between wet and dry days. An anomaly is formally a difference with respect to a long-term average, not a difference between extreme cases.

3. The authors should provide explanation for some of their conclusions which are hard to see from the figures, not shown by the figures, or ignored with respect to the figures:

a. Page 6, Line 3: There is an opposing behavior in this study and Bosilovich et al. (2003) with respect to the change in the recycling contribution from July to September. In this study, this contribution increases, but in Bosilovich et al. (2003), it decreases in time. Can the authors comment on why there is a discrepancy?

b. Page 6, Line 17: The description of 'all the source regions contribute with higher recharges before the synoptic-scale rainfall events' is too vague for a reader to see in the figures. Some regions have higher E-P at different days before the event, with monotonically-increasing, monotonically-decreasing or humped behaviors shown. Perhaps the authors can be more specific as to what days they are comparing?

c. Page 6, Line 31: There is an important difference between wet and dry days for the NAM region on -1 days that is not discussed. A similar difference is noted for EAST on -1 and -2 days. These could potentially be interpreted as evaporation from land surfaces that does not lead to significant rainfall.

d. Page 7, Line 3: The interpretation of difference between wet and dry days of the term E-P is quite difficult for most readers to make. (E-P)wet - (E-P)dry includes four terms, two related to E and two related to P. While the authors cannot separate these terms, it would be useful to explain to the reader why a positive anomaly means rainfall intensification and a negative anomaly means evaporation intensification, if that is the case.

e. Page 7, Line 12: The authors indicate that the NAM region has no systematic change during the wet period, but Figure 9a shows a large change in -2 days. The discussion needs to reflect this difference.

f. Page 7, Line 16: It is hard to see from the figures where the authors have conclusively shown that the northern NAM region shows a particularly important relation to the

fluxes from the Caribbean Sea.

g. Page 8, Line 14: Perhaps the authors need to remove this statement. Other studies have looked at terrestrial land sources over the entire continental land mass in North America. They might not have defined a sub-region called 'southwestern U.S.' in the same way as it is done here, but that is a minor point.

4. The authors have not shown a comparison of their E-P estimates to observational data which limits the credibility of the study. There is a mention of the observational dataset CHIRPS as being used to help specify the lifespan of 6 days through a comparison of simulated precipitation from FLEXPART to CHIRPS. It would be useful to show a comparison of FLEXPART P and CHIRPS P for the NAM region, rather than simply relying on a reference to Perdigon-Morales et al. (2017). Furthermore, the authors are encouraged to compare simulated E or simulated E-P to available observations or other reanalysis products given that the entire validity of this study relies on how well E-P is captured by FLEXPART.

5. The authors have defined precipitation recycling to be a process of local evaporation from the NAM region exceeding precipitation. It is important that the authors be more careful in this definition for a number of reasons:

a. Terrestrial evaporation from other land masses (EAST, NORTH) should also be considered precipitation recycling when it leads to precipitation in the NAM region.

b. Precipitation recycling still occurs even if evaporation does not exceed precipitation. In other words, the authors have equated E>P to recycling, but this is not necessary the case as recycling is still occurring when E<P.

6. The identification and naming of the regions should be reconsidered and perhaps more strongly justified. The names 'NORTH', 'EAST', 'ATL' and 'PAC' could be misinterpreted as these are specific to this study and not generally accepted terms. ATL is Gulf of Mexico/Caribbean Sea, PAC is Baja California and Eastern Pacific, NORTH

is southwestern U.S. and EAST is northeastern Mexico. More insightful and relevant naming would be useful. Most studies in the NAM would also include portions of the NORTH and EAST inside the NAM boundary. Other than justifying their selection by citing Hu and Dominguez (2015), the authors should further explain their choice as it is important for the overall outcomes and interpretation of their work.

Technical Corrections:

Page 1, Line 1: Suggest a change to the title as: "On the origin of atmospheric moisture related to ..."

Page 1, Line 14: Use the term 'precipitation recycling' instead of 'recycling', here and elsewhere in the manuscript.

Page 2, Line 16: The authors could cite Vivoni et al. (2008) who provide estimates of regional fractions of annual precipitation during the monsoon.

Page 3, Line 6: The authors could cite Mendez-Barroso and Vivoni (2010) and Xiang et al. (2018) to help support this statement as these studies specifically look at observed and simulated soil moisture-vegetation-precipitation recycling.

Page 3, Line 30: Replace 'freshwater' with 'water' or 'water vapor', here and elsewhere in the manuscript.

Page 5, Line 2: Geopotential and specific are misspelled.

Page 5, Line 11: The authors can indicate that water vapor gain is E>P, whereas water vapor loss is E<P.

Page 6, Line 18: The term 'water uptake' can be very confusing. Suggest to change to 'water vapor influx'. Correct here and elsewhere in manuscript.

Page 6, Line 18: The term 'recharges' is not used correctly. Suggest to change to 'water vapor influx'.

Figure 1. Please improve this figure. The bathymetry is not required. The NAM region should be labeled. This should include the overall extent of the study domain, including the sub-regions identified later. Additional information is requested: US and Mexico boundary, and state boundaries.

Figure 2. The y-axis should be labeled 'Rainfall events / month'.

Figure 5. NORTH is misspelled.

Figure 6. A legend is needed.

Figure 7. Remove the blue dashed line representing 'zero' as it is confusing.

References:

Mendez-Barroso, L.A., and Vivoni, E.R. 2010. Observed Shifts in Land Surface Conditions during the North American Monsoon: Implications for a Vegetation-Rainfall Feedback Mechanism. Journal of Arid Environments. 74(5): 549-555.

Vivoni, E.R., Moreno, H.A., Mascaro, G., Rodriguez, J.C., Watts, C.J., Garatuza-Payan, J., and Scott, R.L. 2008. Observed Relation between Evapotranspiration and Soil Moisture in the North American Monsoon Region. Geophysical Research Letters. 35: L22403, dos:10.1029/2008GL036001.

Xiang, T., Vivoni, E.R., and Gochis, D.J. 2018. Influence of Initial Soil Moisture and Vegetation Conditions on Monsoon Precipitation Events in Northwest Mexico. Atmosfera. 31(1): 25-45.

---

## Referee Comment (RC2) · A. M. Durán-Quesada (Referee) · 25 Jun 2018

General comments:

The study aims to provide a long term analysis of the sources of atmospheric water vapor for the NAM system and their relationship with synoptic scale rainfall events using a backward Lagrangian trajectories method based upon the FLEXPART mode. Considering moisture supply to the NAM has been mostly analyzed from an Eulerian perspective, the proposed approach provides a new insight to the problem analysis. Moreover, including a more detail analysis in terms of the synoptic rainfall events is regarded as a new and valuable contribution.

[Figure]

Specific comments:

Introduction

1- The introduction condenses a vast amount of previous studies on the NAM, still something can be added to briefly explain why regardless of not fulfilling the wind reversal criteria, the system is considered as a monsoon.

Method

Section 2.1:

Page 3:

2- It is not clear which FLEXPART version was used to generate the trajectories dataset or whether the data was generated for this work at all. Version 9.0 is referred as Stohl et al., 1998 and Stohl and Thomson, 1999 but those correspond to much older versions of the model, version 9.0 was released in 2012. More detail on the dataset generation is needed or mention the correspondent reference of the work for which the data was originally computed that must have the full detail.

Page 4:

3- How does "the difference between simulated precipitation and CHIRPS data" represent a lifetime? Does this refer to a validation of the skills of the trajectories to capture rainy days compared to (I suppose daily) CHIRPS based on a threshold for daily accumulated precipitation and dq/qt?

4- I would suppose CHIRPS is a reasonably good dataset for the analysis domain as a larger amount of observations are included, I would like to recommend some briefing on the accuracy of FLEXPART to capture rainy days compared to CHIRPS to ensure reliability.

Section 2.2:

4- Page 4:

5- Though this section provides an explanation to previous question, the method for selecting events must be better explained.

6- The interpretation of synoptic events is confusing, it is certainly based on a spatial scale considerations which is a bit different to what is expected after reading the title and introduction. Following the title one may expect a full synoptic classification (such as Hochman et al. 2018) that identifies the large scale conditions associated with rainfall for which the (E-P)-n field is analyzed. Instead, the synoptic classification is used to provide a sort of measurement of the precipitation influence area identified following a dry/wet days criteria. A change in the title is suggested to avoid confusion in what to expect from the method and results.

7- Caption of figure 3 is required to be self-explanatory. The cut in the figure looks weird, you can use a larger domain to show a more complete map and contour only the region of interest.

Results

Section 3.1:

8- There seems to be a misunderstanding on the interpretation of (E-P)-n (when using trajectories analysis) for it is known that the presence of a moisture source is valid interpretation for (E-P)-n > 0 over ocean, the accuracy of a similar interpretation for (E-P)-n > 0 over land is rather questionable (the bias of E estimation is high with this method so that interpretation of recycling ratios is restricted to an upper level, see full detail in Stohl and James, 2004). Further aspects are to be considered for recycling, namely a few: a) meaning of the "E" term and how does it reflect or not recycling processes such as transpiration, b) the scale dependency of moisture recycling rations need to be taken into consideration (see e.g Van der Ent and Savenije, 2011) , c) what

does "recycling" actually mean in terms of the modeled scales?

9- The identification of the "five moisture sources" is then limited to the interpretation of the (E-P)-n estimates. Hence, strictly speaking, the results for sources identified as NAM, EAST and NORTH need a through review for the full manuscript. Same for every result related to inferred "recycling".

10- The GoC has been previously highlighted as a relevant source of moisture for the NAM development, however the results show that it is not the GoC but the region off the Pacific coast which acts as a moisture supplier. Does this present a contrasting result compared to previous work? How do you interpret the result in comparison with previous works?

11- figures 5 and 6 should be modified according to the considerations of the "precipitation recycling" interpretation.

Previous comments on interpretation of precipitation recycling apply for this section as well the full document.

12- [Considering the oceanic sources only] results for the difference among weak, moderate and extreme rainfall events (figure 9) show very little variations from one case to the other. How do you interpret this result in terms of moisture availability, transport and observed precipitation for the events? The use of other variables such as precipitable water vapor and a measure of atmospheric stability could provide support for analysis.

13- Vectors in figure 10 are not easy to ready, you can try plotting them every 5 or 10 grid points to improve the figure. The discussion regarding the interpretation of the Geopotential height and moisture transport anomalies in the analysis in page 7needs improvement. Consider for example discussing the dynamics underlying the large scale patterns and the bin of event (weak, moderate, extreme).

Summary and concluding remarks

14- Considering the authors have defined the recycling of precipitation in terms of local evaporation over the NAM domain, the analysis and this section need a revision. It is key to note that the time scale of the simulations does not necessarily fit the scale of processes that occur at local scales.

References

- Hochman A, Harpaz T, Saaroni H, Alpert P. Synoptic classification in 21st century CMIP5 predictions over the Eastern Mediterranean with focus on cyclones. International Journal of Climatology. 2018 Mar 1;38(3):1476-83.

- Van der Ent, R.J. and Savenije, H.H.G., 2011. Length and time scales of atmospheric moisture recycling. Atmospheric Chemistry and Physics, 11(5), pp.1853-1863.

---

## Short Comment (SC1) · 10 Jul 2018

Review of Ordoñez et al. 2018 David K. Adams (dave.k.adams@gmail.com)

I commend the authors, several of whom are my colleagues, efforts to attack this important problem, a problem that goes way beyond the scope of the North American Monsoon region. However, there are some fundamental problems with this paper that need to be addressed before publication. It is clear the study is fundamentally flawed by the conclusion that the Pacific Ocean off the coast of California is a moisture source for the monsoon. This is patently absurd given the water temperatures (10 to 20C) of the Pacific Ocean (see typical sounding Figure 2 below), no need to even consider the

stable Subtropical Pacific High and nearly impassable mountains of California. So, just for this reason alone, all other results from this study should be called into question. However, the indication that the Sonoran and Mojave Desert could be a major moisture source through also stretches the limits of credibility– one of the driest, warmest regions on Earth that receives scares, very localized rainfall during the Monsoon becoming an important local source. These results are surely a result of the FLEXPART modeling employed which is apparently inadequate for the task at hand.

These radical, game-changing conclusions that the authors arrive at, contradicting nearly 4 decades of the vast majority of studies (both observational and modeling), need to provide extraordinary evidence (particularly direct observational support, not modeled products of water vapor or precipitation products over Mexico) for their conclusions.

Below I outline basic points that should be addressed in order to validate their conclusions.

(1) From simple thermodynamic arguments (the water vapor scale height is only about 2km), mid-level cannot be responsible for for the in the NAM region west of the elevated terrain (SMO, Continental Divide, etc). This is shown by Adams and Souza (2009), Maddox et al. 1995, Mazon et al. 2016, Rogers et al. 2017 and many, many others. Low-level water vapor (below 800mb) is required for generating convective instability over the valleys and low deserts. Elevated water vapor sources (above $\sim 700$mb) can help with initiation over elevated regions (e.g. SMO or Four-Corners region) but this convection cannot migrate towards lower regions without the support of low-level moisture. Entrainment of dry air is too detrimental. Gulf of Mexico can certainly contribute to NAM convective precipitation, but can not "cause" or be "responsible" for a sizeable portion of it, west of the SMO/Continental Divide regions. No low-level moisture, no deep convective precipitation in low-lying zones. Likewise, no strong surface heating, no deep convection (higher level moisture can actually be detrimental to the later, soundings can be too moist ( see Adams and Souza 2009))

(2) The Model. The authors will have to provide more evidence as to how their extremely simplistic model (FLEXPART) which cannot account for vertical transport in convective updrafts, (scale 100s of meters to kms on the high end of organized convection, and order minutes to an hour), nor for entrainment/mixing in a realistic manner, and no need to mention the proper formation of hydrometeors and precipitation processes. Even assuming FLEXPART is a valid model for doing the impossible, consider also the arguments of Ana Maria Quesada. The evaporation/precipitation relationship is entirely space/time resolution dependent (See our figure 3 below, local surface fluxes have little relationship with total column, precipitable water vapor).

(3) Data. ERA-Interim data is not adequate. Reanalysis data, in general, is inadequate because model-generated values depend on convective and microphysical parameterizations. Pressure, geopotential heights and winds, can be dynamically constrained, the water vapor distribution cannot. This poorly measured quantity is extremely difficult to replicate in the NAM region (see Radhakrishna et al. 2016 and many others) (Also, See attached figure 3 for ERA-Interim vs GPS Precipitable water vapor data below). ERA-5 is still bad, but much better than ERA-Interim or NARR. Given the radically different nature of surface evaporation of the oceanic (e.g., different surface wind speeds and temperatures) zones and more critically over the complex topography of Mexico and the Southwest U.S., how is the evaporation determined at the necessary spatial/temporal resolution.

(4) Moisture recycling Just a point of logic, if local moisture recycling is important then (1) how does the monsoon precipitation begin? Vegetation green up occurs at the end of July into August (2) Why does precipitation decrease, become for more variable around the second week of August (see for example Kursinski et al. 2008) in Northwestern Mexico and Arizona even though vegetation is green and surface moister, in general? Also, wet surfaces can actually be detrimental to deep convection, as convective temperatures cannot be reached for this region (see Kirsten Findell's 2011 article among others). Our results from the GPS Hydromet Network show that at the subdiurnal scale, local water vapor fluxes and precipitable water vapor have small correlations, lagged or not. So at least locally, it is not apparent that moisture recycling (see figure 3 below) plays any important role in column water vapor. Boundary layers are extremely deep and well-mixed over much of the region, so very local surface evaporation would be mixed out quickly, not providing sufficient "density" to either contribute to increasing convective instability, nor precipitation efficiency. Only large-scale advection could account for generating the necessary water vapor fluxes to produce the instability necessary to generate convective activity.

5) And probably most important and what makes this a very difficult problem to separate sources easily is the mixing over the SMO due to deep convective activity. Moist air to the east of Arizona and the SMO, in general, may result from deep convective mixing further south and then transported northward along the high terrain with or without precipitation.

Minor Comments

"Later studies claimed" is not the correct word, better "indicated" or something of the like. The vast majority of studies over the last several decades have shown that the time mean as well as transient flow are dominated by the EPac and GoC.

"While compelling, these results are based on observations from a limited field campaign" These are a least real-world data and not model dependent as are ERA-Interim.

"IVs, as well as improved models of the flow over complex terrain like that of the NAM region are both needed to better understand the role of IVs in supporting convective outbreaks across the monsoon region." This is sort of a throwaway statement. Our understanding of IV is fairly good and there dynamic effects (increased shear) appears to be responsible for convective organization. Water vapor transport plays no role, nor its advection, as these features are found at 300 or 200mb. (See Finch and Johnson 2010).

IVs seem to help the moist air to bypass the Sierra Madre westward accompanied by organized convection and upward vertical velocity over the NAM region. You need to back this claim up. As shown by Finch and Johnson (2010). windshear in organizing convection is important, adiabatic lifting is weak. From our examination of Marty Ralph and Tom Galarneau's results (using lightning data as a proxy) Invited Manuscript for Atmospheric Research there is little evidence of low-level water vapor transport from the Gulf of Mexico on strong convective days. Strong shear can certainly be important in the easterly winds, but we should not convolve dynamics and water vapor transport.

"This wind reversal is not of sufficient magnitude and scale to meet the criteria of Ramage (1971) for a monsoon (Hoell et al. 2016)." You need more the two citations to make the claim that it is not a monsoon circulation.

"Bosilovich et al. (2003) found the dominant sources of monsoon precipitation to be the local evaporation and transport from the tropical Atlantic Ocean (including the GOM and Caribbean Sea)." As Bosilovich et al. (2003) note in their study "It is also worthwhile to reiterate that the model does not resolve the Gulf of California, which should influence the sources of water." Always critical to consider space/time resolution of large-scale models.

Our work with Chris Castro and others demonstrates the need for Convective Resolving Models in order to capture mesoscale convective systems (order kms resolution), which are responsible for a large portion of precipitation particularly in NW Mexico. Results from low resolution models 1degree x 1 degree should be critically assessed given their inability to produce these systems correctly. (see Lamhers et al. 2016, Luong et al. 2017, Moker et al in press JAMC)

Figure 1. (Figures 1,2,3 in comments) Percent error between ERA-Interim PWV and collocated GPS PWV from NAM GPS Transect Experiment 2013 (See Serra et al. 2016). Errors for most of the 10 GPSmet sites can exceed 20%.

Figure 2) (Figure 4 in comments) Typically July sounding for Oakland and San Diego.

Low Moist Static Energy at surface could simply not provide sufficient convective instability and could, under no conditions, break through the extreme inversion always present over the Pacific Ocean from July to mid-September.

Figure 3. (Figure 5 and 6 in comments) Latent Heat Flux vs GPS PWV for Rayón Sonora (top) during active Monsoon period.(Bottom) Correlation of above LH flux vs PWV figure smooth for 3 hours. Correlations with different smoothing are typically 0.3 or less on diurnal to sub-diurnal timescales. Data from NAM GPS Hydromet Experiment 2017.

[Figure]

**Fig. 1.**

**PWV Error ERA Interim vs ONVS GPS Transect 2013**

Percent Error (%) vs Decimal Day

**Fig. 2.**

[Figure]

Fig. 3.

[Figure]

**Fig. 4.**

[Figure]

**Fig. 5.**

[Figure]

**Fig. 6.**

---

## Author Comment (AC1) · 9 Aug 2018

**Responses to Dr. E. R. Vivoni (Referee comments)**

**General Comments:**

The study describes the sources of atmospheric water vapor during the North American monsoon and its dependence on the size of synoptic-scale rainfall periods. To achieve this, a particle tracking analysis of air parcels was applied using a reanalysis product in order to estimate fields of evapotranspiration minus precipitation (E - P) taken as the metric to study sources and sinks of water vapor. The authors have done a very nice job in bringing a different tool to a subject matter that is under considerable debate. Their work provides some new insights that will be useful for the community at large to consider. Furthermore, their work is well written and illustrated. The comments below are intended for the authors to improve their work and increase its overall impact by demonstrating the solid nature of their method, results and interpretations.

We thank the reviewer for his kind comments that will help to improve the manuscript. We will revise the paper according to the comments. Responses to the specific comments are given below.

**Specific Comments:**

1. It is important that the authors have a more in-depth explanation of the FLEXPART algorithm as it pertains to the generation of the E-P field, including the limitations therein, within the introduction and methods section. Some of the limitations are indicated subsequently in the results section, which is considered too late in the manuscript. What type of errors in E-P are expected from its estimation as dq/dt?.
How sensitive is the method to the selection of the ERA-Interim reanalysis fields at 1 degree, 3 hourly resolution?
How realistic are these fields with respect to observations or other reanalysis products that independently estimate P and E, and from which E-P can be obtained? The reader needs to have confidence in the accuracy of E-P before it is used to make inferences on the sources and sinks of water vapor in the monsoon region.

(E - P) diagnosed by FLEXPART has been already compared with (E - P) obtained using other data that individually estimate E and P with similar results (Stohl and James, 2004), so we are quite confident in the methodology. However, we agree with the reviewer and in our paper we do not explain this too convincingly. In this regard, the excellent papers by Gimeno et al. (2010) or Gimeno et al. (2012) contains a detailed explanation of the methodology and its reach, by using the very same kind of data as us (resolution, source of data, etc.). In the revised version, we will refer the reader to these works for a revision in depth of the methods.

Notwithstanding, we will explain the FLEXPART-based method to calculate E-P more in depth including a subsection in the "method section" explaining the limitations of the Lagrangian approach.

In our work, we used the tracks computed using ECMWF available data: operational analyses available every 6 hours (00, 06, 12 and 18 UTC) plus short-term forecasts available at intermediate times (3, 9, 15, 21 UTC). We will obtain (E - P) from a combination of observed E and P datasets (eg. E from GLEAM and P from CHIRPS for terrestrial surfaces, and E from OAFLUX and P from GPCP for oceanic surfaces, although the latter only cover roughly a half of the study period) and by computing the divergent part of the water vapour flux (Eulerian method) from other reanalysis (as CFSR). We will compare these results with the figure 4 of the manuscript.

2. It would be useful for the authors to present a justification and/or further detailed explanation for the following aspects of their methodology:

Modifications will be incorporated following these suggestions. Please, see details below.

a. The coarse resolution (1 degree by 1 degree) of the meteorological fields used in the FLEXPART model, given the scale of land and ocean features in the NAM region.

In this work, we are obtaining average values from a climatological perspective. At this scale, the $1^\circ$x$1^\circ$ resolution is sufficient to capture the moisture origin of a large area. For this kind of analysis, we run FLEXPART (v9 experiment in this case) on a global domain. In our case, the atmosphere was divided into approximately 2 million homogeneous particles. Running global-scale FLEXPART experiments at finer scales would most probably not affect the results (at the scale we deal with in our research) and it would be extraordinarily time consuming, and impossible in terms of computing power for our team.

b. The selection of the time period (1981-2014).

1981 is the first year when CHIRPS data exist. When we started the study, 2014 was the last available complete year of ERA-Interim for running FLEXPART. In this work, we have a 34-yr climatology, we consider that adding two or three more years wouldn't be a significant difference. We will explicitly explain our time period in the revised version.

c. The boundary selected to represent the NAM region and its consistency with the tiered approach advocated by Higgins and co-authors during the North American Monsoon Experiment.

In this work, our objective is to study the NAM as a monsoon, i.e. at a broad scale, to detect long-range atmospheric water vapor transport that could influence the precipitation. Nevertheless, the core of the NAM region is based on the North American Monsoon Experiment and in consequence, it is consistent with it. We will explicitly include this information in the revised paper.

d. The use of the term anomaly when discussing differences between wet and dry days. An anomaly is formally a difference with respect to a long-term average, not a difference between extreme cases.

The referee is right. The term "differences" instead of "anomalies" is more precise in this case. We will change it.

3. The authors should provide explanation for some of their conclusions which are hard to see from the figures, not shown by the figures, or ignored with respect to the figures:

We will revise the text in order to further detail our explanations, in particular in relation to the figures.

a. Page 6, Line 3: There is an opposing behavior in this study and Bosilovich et al. (2003) with respect to the change in the recycling contribution from July to September. In this study, this contribution increases, but in Bosilovich et al. (2003), it decreases in time. Can the authors comment on why there is a discrepancy?.

Strictly, we are not computing the recycling, but E-P over the NAM region itself, and in this sense, as the other reviewer also suggests, the term recycling can be confusing and we will change it.

Additionally, our approach is not strictly comparable to that of Bosilovich et al. First, these authors compute the fraction of precipitation that originates as a result of the evaporation from the region. Unfortunately, we can't separate E and P with our methodology. Second, in this figure, we are not computing E-P before precipitation events but the general climatology of E-P. We commented it briefly in lines 9-11 of page 6.

However, this is an interesting question. In the revised version we will specifically address this comparison as best as we can with our approach. We will include a new figure for E-P before precipitation events to go deeper into this question.

b. Page 6, Line 17: The description of 'all the source regions contribute with higher recharges before the synoptic-scale rainfall events' is too vague for a reader to see in the figures. Some regions have higher E-P at different days

before the event, with monotonically-increasing, monotonically-decreasing or humped behaviors shown. Perhaps the authors can be more specific as to what days they are comparing?.

In the revised version we will improve (and expand) this explanation.

c. Page 6, Line 31: There is an important difference between wet and dry days for the NAM region on -1 days that is not discussed. A similar difference is noted for EAST on -1 and -2 days. These could potentially be interpreted as evaporation from land surfaces that does not lead to significant rainfall.

It could be interpreted for the NAM region because in the case of NAM during the day -1 before the rainfall events E-P>0, therefore evaporation still dominates.

In the case of the EAST region E-P<0, so the region is a net sink of moisture during days -2 and -1. We interpret that the particles over EAST are already losing moisture during these days because this region is close to the target region (NAM).

We will explicitly address these differences and their implications in the revised version.

d. Page 7, Line 3: The interpretation of difference between wet and dry days of the term E-P is quite difficult for most readers to make. (E-P)wet - (E-P)dry includes four terms, two related to E and two related to P. While the authors cannot separate these terms, it would be useful to explain to the reader why a positive anomaly means rainfall intensification and a negative anomaly means evaporation intensification, if that is the case.

An explanation will be included in the revised version.

e. Page 7, Line 12: The authors indicate that the NAM region has no systematic change during the wet period, but Figure 9a shows a large change in -2 days. The discussion needs to reflect this difference.

It is difficult to interpret, but E-P before the weak precipitation days presents a behaviour similar to dry days. In contrast, moderate and heavy rainfall events show another completely different behaviour that could be related to the strong surface heating that is needed prior to these kind of precipitation events. This will be argued in the text.

f. Page 7, Line 16: It is hard to see from the figures where the authors have conclusively shown that the northern NAM region shows a particularly important relation to the fluxes from the Caribbean Sea.

Figure 3 shows that the main difference between the pattern of moderate and extreme rainfall intensity is found over the northern NAM. In figure 8a we can also see that main differences of E-P at day -1 are over the northern NAM. However, we concur with the referee and it is difficult to appreciate these changes. We will clarify them in the revised version.

g. Page 8, Line 14: Perhaps the authors need to remove this statement. Other studies have looked at terrestrial land sources over the entire continental land mass in North America. They might not have defined a sub-region called 'southwestern U.S.' in the same way as it is done here, but that is a minor point.

We will take into account this comment in the revised version.

4. The authors have not shown a comparison of their E-P estimates to observational data which limits the credibility of the study. There is a mention of the observational dataset CHIRPS as being used to help specify the lifespan of 6 days through a comparison of simulated precipitation from FLEXPART to CHIRPS. It would be useful to show a comparison of FLEXPART P and CHIRPS P or the NAM region, rather than simply relying on a reference to Perdigon-Morales et al. (2017).
Furthermore, the authors are encouraged to compare simulated E or simulated E-P to available observations or other reanalysis products given that the entire validity of this study relies on how well E-P is captured by FLEXPART.

The above-mentioned lifespan was computed both with CHIRPS and ERA-Interim precipitation data with the same result of 6 days.

We use CHIRPS data to classify the precipitation events because we have confidence that CHIRPS data are more realistic than ERA-Interim precipitation database. CHIRPS incorporates 0.05° resolution satellite imagery with in-situ station data, meanwhile ERA-Interim precipitation data comes from the atmospheric model forecast.

Besides that CHIRPS have been proved by Perdigón-Morales (2017) to reproduce properly some of the particular characteristics of the Mexican rainfall as the Mid-Summer Drought, we also performed a test founding that CHIRPS precipitation classification is consistent with the classification from the observed PWV obtained by GPS network over three locations over the NAM (Figure R1).

[Figure]

**Figure R1**. Precipitable water vapor average during dry days (black line), weak rainfall days (light blue line), moderate rainfall days (dark blue) and extreme rainfall days (purple line) at (a) AZCO, (b) SA48, (c) HER2 (locations are indicated in the map below) for the years 2013/2014.

[Figure]

FLEXPART doesn't use precipitation data from ERA-Interim to estimate (E-P) but "q", "u", "v", "w" which comes from the atmospheric model analysis and not from the atmospheric model forecast. We use the ERA-Interim reanalysis because this dataset has been found to provide a reliable representation of the atmospheric component of the hydrological cycle (Lorenz and Kunstmann 2012; Trenberth et al. 2011).

As we mentioned in the comment #1, we will compare the simulated (E - P) with (E - P) from a combination of observed E and P datasets and with (E - P) obtained by the Eulerian method.

If we ignore the presence of liquid water and ice in the atmosphere, the water budget in an atmospheric column can be written as:

$$E - P = \frac{\partial w}{\partial t} + \nabla \frac{1}{g} \int_0^{p_s} q v \, dp$$

Where E-P is the surface freshwater flux, $w = 1/g \int_0^{p_s} q \, dp$ is the precipitable water, "t" is time, "g" is the gravitational acceleration, "q" is the specific humidity, "v" is the wind, and "E" is the evaporation and "P" the precipitation rate per unit area (Trenberth and Gillemot, 1998).

Averaged over time, the rate of change of water storage is small, and E-P is largely balanced by the second term of the right hand. Satellite observations with fine temporal and spatial resolution have shown great promise in improving the estimation of this term (Gimeno et al., 2012).

5. The authors have defined precipitation recycling to be a process of local evaporation from the NAM region exceeding precipitation. It is important that the authors be more careful in this definition for a number of reasons:

a. Terrestrial evaporation from other land masses (EAST, NORTH) should also be considered precipitation recycling when it leads to precipitation in the NAM region.
b. Precipitation recycling still occurs even if evaporation does not exceed precipitation. In other words, the authors have equated E>P to recycling, but this is not necessary the case as recycling is still occurring when E<P.

We have approximated the E-P budget over the NAM by the recycling process. As the referee points out, this is not strictly correct, we will substitute the term "recycling" by an "evaporative source form de NAM region itself".

6. The identification and naming of the regions should be reconsidered and perhaps more strongly justified. The names 'NORTH', 'EAST', 'ATL' and 'PAC' could be misinterpreted as these are specific to this study and not generally accepted terms. ATL is Gulf of Mexico/Caribbean Sea, PAC is Baja California and Eastern Pacific, NORTH is southwestern U.S. and EAST is northeastern Mexico. More insightful and relevant naming would be useful. Most studies in the NAM would also include portions of the NORTH and EAST inside the NAM boundary. Other than justifying their selection by citing Hu and Dominguez (2015), the authors should further explain their choice as it is important for the overall outcomes and interpretation of their work.

The names of the regions will be changed to be consistent to previous studies as follows:

NAM ----->NAM
PAC ----->NEP
PAC ----->GOC
ATL ----->GOM-CAR
NORTH ----->SW-US
EAST ----->NE-MEX

Please note that the region PAC has been divided in two regions, GOC and the North Eastern Pacific coast (NEP) as suggested by the reviewer Dra. Duran-Quesada. Please see her comment # 10.

We chose the same boundaries than Hu and Dominguez (2015) because they used a semi-Lagrangian scheme (DRM) to delineate and quantify the climatological moisture sources of NAM precipitation. In this work, we also determine the climatological moisture sources of NAM with a similar but more sophisticated three-dimensional Lagrangian method. As we are working in the same space and time scales we made the results more directly comparable by choosing the same domain. However, it is important to note that the choice of Hu and Dominguez (2015) was based on both the North American Monsoon Experiment (NAME) Science and Implementation Plan and NAME precipitation zones, as defined by Castro et al. (2012).

**Technical Corrections:**

Corrections will be incorporate following the suggestions of the reviewer.

**References**

Castro, C.L., Chang, H.I. and Dominguez, F. 2012: Can a regional climate model improve the ability to forecast the North American monsoon? *J. Climate*, 25, 8212–8237, doi: 10.1175/ JCLI-D-11-00441.1

Gimeno, L., A. Drumond, R. Nieto, R. M. Trigo, and A. Stohl, 2010: On the origin of continental precipitation. *Geophys. Res. Lett.*, 37, L13804, Doi: 10.1029/2010GL043712.

Gimeno, L., Stohl, A., Trigo, R.M., Dominguez, F., Yoshimura, K., Drumond, A., Duran-Quesada, A.M., Niero, R. 2012: Oceanic and terrestrial sources of continental precipitation. *Reviews of Geophysics*, 50, RG4003. Doi: 10.1029/2012RG000389

Hu, H. and Dominguez, F. 2015: Evaluation of oceanic and terrestrial sources of moisture for the North American monsoon using numerical models and precipitation stable isotopes. *J. Hydrometeor.*, 16: 19–35, doi:10.1175/JHM-D-14-0073.1, 2015.

Lorenz C, Kunstmann H. 2012: The hydrological cycle in three state of-the-art reanalyses: Intercomparison and performance analysis. *J Hydrometeor* 13(5):1397–1420. doi:10.1175/jhm-d-11-088.1

Stohl, A. and James, P. 2004: A Lagrangian analysis of the atmospheric branch of the global 5 water cycle. Part 1: Method description, validation, and demonstration for the August 2002 flooding in central Europe, *J. Hydrometeorol.*, 5: 656– 678, 2004.

Trenberth K.E., Fasullo J.T., Mackaro J. 2011: Atmospheric moisture transports from ocean to land and global energy flows in reanalyses. *J Climate* 24(18):4907–4924. doi:10.1175/2011jcli4171.1

Trenberth, K.E., and C. J. Guillemot 1998: Evaluation of the atmospheric moisture and hydrological cycle in the NCEP/NCAR reanalysis, *Clim. Dyn.*, 14, 213–231, doi:10.1007/s003820050219.

---

## Author Comment (AC2) · 9 Aug 2018

**Responses to Dr. A. M. Durán-Quesada (Referee comments)**

**General comments:**

The study aims to provide a long term analysis of the sources of atmospheric water vapor for the NAM system and their relationship with synoptic scale rainfall events using a backward Lagrangian trajectories method based upon the FLEXPART mode. Considering moisture supply to the NAM has been mostly analyzed from an Eulerian perspective, the proposed approach provides a new insight to the problem analysis. Moreover, including a more detail analysis in terms of the synoptic rainfall events is regarded as a new and valuable contribution.

We thank the reviewer for the thoughtful comments that will improve the manuscript. Please find below the response to the specific comments.

**Specific comments:**

Introduction

1- The introduction condenses a vast amount of previous studies on the NAM, still something can be added to briefly explain why regardless of not fulfilling the wind reversal criteria, the system is considered as a monsoon.

The pioneering studies using the reversal in wind direction to identify a monsoon domain were designed to work for the Eastern Hemisphere (Ramage, 1971). Because the seasonal wind reversal is less significant over the Americas, this criterion was subsequently revised in order to consider the characteristics of the monsoonal precipitation.

Monsoonal precipitation is characterized by a concentration of yearly rainfall in the local summer and a dry period in the local winter. Using simple parameters based on precipitation, the global monsoon area have been more recently redefined (e.g. Wang and Ding, 2008; Wang et al., 2012; Huo-Po and Jia-Qi, 2013; Lee and Wang, 2014; Liu et al., 2016; Mohtadi et al., 2016; Wang et al., 2017; Wang et al., 2018). In this approach, both the North America Monsoon System (NAMS) and the South America Monsoon System (SAMS) are clearly found (figure R1):

However, the term "NAM" has also been extensively used to refer to the precipitation of Sinaloa Sonora and Southern Arizona (in northwestern Mexico and southwestern U.S.). We adopt the latter definition of the NAM, but in this manuscript, we pretend to briefly reflect on the suitability of using the same name in the existing scientific literature to refer to climatological features of different regions.

In the revised version we will explicitly include a discussion on this important comment.

[Figure]

**Figure R1.** Regional land monsoon domain based on 26 CMIP5 multi-model mean precipitation with a common 2.5° × 2.5° grid in the present-day (1986-2005). For regional divisions, the equator separates the northern monsoon domains from the southern monsoon domains. All the regional domains are within 40°S to 40°N. Source: IPCC, 2013.

Method

Section 2.1

Page 3:

2- It is not clear which FLEXPART version was used to generate the trajectories dataset or whether the data was generated for this work at all. Version 9.0 is referred as Stohl et al., 1998 and Stohl and Thomson, 1999 but those correspond to much older versions of the model, version 9.0 was released in 2012. More detail on the dataset generation is needed or mention the correspondent reference of the work for which the data was originally computed that must have the full detail.

We have used FLEXPART v9. We will clarify this in the revised version including an update of the references.

Page 4:

3.- How does "the difference between simulated precipitation and CHIRPS data" represent a lifetime? Does this refer to a validation of the skills of the trajectories to capture rainy days compared to (I suppose daily) CHIRPS based on a threshold for daily accumulated precipitation and dq/qt?

The time period was calculated following this methodology:

1.- First the sources of moisture where calculated (using the backward mode) for the NAM region integrating E-P>0 values during 10 days.

2.- Next, from the complete sources of moisture, defined in point 1, we calculate E-P<0 (P-FLEX) using the backward mode for 1 day to 10 days, individually.
3.- Then, we calculate E-P<0 over each grid over NAM region for each integrated time. So, we have 10 P-FLEX values for each grid point.
4.- We calculated the best "integrated time" for each grid point as the difference between CHIRPS data and each P-FLEX.
5.- Finally, we obtain a "mean time" out of these "individual grid" values.

As a result, we detect that the best time to reproduce the precipitation with FLEXPART over the NAM region is 6 days.

This procedure will be clarified in the revised version of the paper.

4- I would suppose CHIRPS is a reasonably good dataset for the analysis domain as a larger amount of observations are included, I would like to recommend some briefing on the accuracy of FLEXPART to capture rainy days compared to CHIRPS to ensure reliability.

We will compute the precipitation modelled by FLEXPART during the last time step of the trajectories to assess how FLEXPART represents precipitation at this stage.

Section 2.2:

Page 4:

5- Though this section provides an explanation to previous question, the method for selecting events must be better explained.

We will explain with further detail.

6- The interpretation of synoptic events is confusing, it is certainly based on a spatial scale considerations which is a bit different to what is expected after reading the title and introduction. Following the title one may expect a full synoptic classification (such as Hochman et al., 2018) that identifies the large scale conditions associated with rainfall for which the (E-P)-n field is analyzed. Instead, the synoptic classification is used to provide a sort of measurement of the precipitation influence area identified following a dry/wet days criteria. A change in the title is suggested to avoid confusion in what to expect from the method and results.

We agree with the reviewer, a new title is proposed, something like "Climatological moisture sources for the North American Monsoon through a Lagrangian approach: their influence on precipitation intensity".

7- Caption of figure 3 is required to be self-explanatory. The cut in the figure looks weird, you can use a larger domain to show a more complete map and contour only the region of interest.

Thanks. We will improve figure 3 in the revised version.

Results

Section 3.1:

8- There seems to be a misunderstanding on the interpretation of (E-P)-n (when using trajectories analysis) for it is known that the presence of a moisture source is valid interpretation for (E-P)-n > 0 over ocean, the accuracy of a similar interpretation for (E-P)-n > 0 over land is rather questionable (the bias of E estimation is high with this method so that interpretation of recycling ratios is restricted to an upper level, see full detail in Stohl and James, 2004). Further aspects are to be considered for recycling, namely a few: a) meaning of the "E" term and how does it reflector not recycling processes such as transpiration, b) the scale dependency of moisture recycling rations need to be taken into consideration (see e.g Van der Ent and Savenije, 2011), c) what does "recycling" actually mean in terms of the modeled scales?.

We understand the reviewer's concern. We roughly approximated (E-P)>0 over the NAM itself to the recycling ratio but as the referee points out, this is not exactly correct (see also our response to the other referee). We will change the term "recycling" by "inland evaporative sources".

9- The identification of the "five moisture sources" is then limited to the interpretation of the (E-P)-n estimates. Hence, strictly speaking, the results for sources identified as NAM, EAST and NORTH need a through review for the full manuscript. Same for every result related to inferred "recycling".

We will revise and further explain our interpretation of the NAM, EAST and NORTH sources. In particular, we will make this discussion consequent with the changes proposed in comment #8.

10- The GoC has been previously highlighted as a relevant source of moisture for the NAM development, however the results show that it is not the GoC but the region off the Pacific coast which acts as a moisture supplier. Does this present a contrasting result compared to previous work? How do you interpret the result in comparison with previous works?.

In fact, the main moisture source over the Pacific is the GoC area. We joined the entire Pacific coast as a moisture source region for simplicity because of the homogeneity of the mean seasonal flow at lower levels showed in figure R2. We performed some sensitive tests that proved that the GoC area is indeed the main moisture source and that the results for the complete region could be compared to previous works. However, we will separate the results for the GoC and the Eastern North Pacific coast in the revised manuscript.

[Figure]

**Figure R2.** Average horizontal wind at sigma 995 for the period 2000-2010.

11- Figures 5 and 6 should be modified according to the considerations of the "precipitation recycling" interpretation. Previous comments on interpretation of precipitation recycling apply for this section as well the full document.

We will clarify this. Please, see our response to comment #8.

12- [Considering the oceanic sources only] results for the difference among weak, moderate and extreme rainfall events (figure 9) show very little variations from one case to the other. How do you interpret this result in terms of moisture availability, transport and observed precipitation for the events? The use of other variables such as precipitable water vapor and a measure of atmospheric stability could provide support for analysis.

In figure 9 an apparent increase of moisture is seen over northeastern Mexico, that extends along part of the Gulf of Mexico and the Caribbean Sea. We interpret this result as the signature of larger moisture transport toward the NAM during the days leading to extreme precipitation events than those leading to low

precipitation events. To add confidence to this result, we will compute the statistical significance associated to these differences for the revised version of this work.

We speculate that this moisture transport could be related to the development of low-level troughs or upper levels IVs that enhance precipitation over the northern NAM. Moisture from large-scale patterns can influence pre-surge air masses and thus surge generation. However, regarding the IVs we allowed ourselves to be a little bit more speculative, since the moisture processes related to the development of IVs are still an open question. This research will be carried out in the future but we consider that it is out of the scope of this paper. We concur with the referee in this point, and it is important to explicitly stablish that this interpretation is speculative and not fully developed in this paper.

One possible limitation of many numerically based climatological moisture source studies of precipitation is the lack of observational data for validating the results. We will compute composites of PWV during the days before of the weak, moderate and extreme precipitation events over the GOC and over the GOM-Caribbean Sea for supporting our result.

13- Vectors in figure 10 are not easy to ready, you can try plotting them every 5 or 10 gridpoints to improve the figure. The discussion regarding the interpretation of the Geopotential height and moisture transport anomalies in the analysis in page 7 needs improvement. Consider for example discussing the dynamics underlying the large scale patterns and the bin of event (weak, moderate, extreme).

We will improve figure 10 in the revised version. A deeper discussion in the interpretation of the Geopotential height and moisture transport anomalies will be included.

Summary and concluding remarks

14- Summary and concluding remarks. Considering the authors have defined the recycling of precipitation in terms of local evaporation over the NAM domain, the analysis and this section need a revision. It is key to note that the time scale of the simulations does not necessarily fit the scale of processes that occur at local scales.

We absolutely concur with the referee at his point. Time scale of the simulations fits the monsoonal scale processes. We are not (in fact we cannot) evaluating processes that occur at local scale. In fact, this is the reason why the precipitation events are detected at a synoptic scale, being representative of a large area of

the studied region. We will try to be sure that this essential fact is clear throughout the revised manuscript and particularly in the final conclusions.

**References**

Huo-Po, C. Jian-Qi, S. 2013. How Large Precipitation Changes over Global Monsoon Regions by CMIP5 Models?, Atmospheric and Oceanic Science Letters, 6:5, 306-311.

IPCC, 2013: Climate Phenomena and their Relevance for Future Regional Climate Change. In: Climate Change 2013: The Physical Science Basis. Contribution of Working Group I to the Fifth Assessment Report of the Intergovernmental Panel on Climate Change [Stocker, T.F., D. Qin, G.-K. Plattner, M. Tignor, S.K. Allen, J. Boschung, A. Nauels, Y. Xia, V. Bex and P.M. Midgley (eds.)]. Cambridge University Press, Cambridge, United Kingdom and New York, NY, USA.

Lee, J-Y, Wang, B. 2014. Future change of global monsoon in the CMIP5. Clim Dyn 42:101–119. DOI 10.1007/s00382-012-1564-0

Liu, F., Chai, J., Wang, B., Liu, J., Zhang, X., Wang, Z. 2016. Global monsoon precipitation responses to large volcanic eruptions. Nature Scientific Reports , 6: 24331. DOI: 10.1038/srep24331

Mohtadi, M., Prange, M., Steinke, S. 2016. Palaeoclimatic insights into forcing and response of monsoon rainfall. Nature volume 533, pages 191–199. doi:10.1038/nature17450

Ramage, C.S.: Monsoon meteorology. Academic Press, London, p 296, 1971

Wang, B., Ding, Q., 2008. Global monsoon: dominant mode of annual variation in the tropics. Dyn. Atmos. Oceans 44, 165–183.

Wang, B., Liu, J., Kim, H.J., Webster, P.J., Yim, S-Y. 2012. Recent change of the global monsoon precipitation (1979–2008). *Clim Dyn* 39:1123–1135. DOI 10.1007/s00382-011-1266-z

Wang, P.X., Wang, B., Cheng, H., Fasullo, J., Guo, ZT, Kiefer, T., Liu; ZY. 2017. The global monsoon across time scales: Mechanisms and outstanding issues. Earth-Sciences Reviews, 174: 84-121.

Wang, B., Li, J., Cane, M.A., Liu; J., Webster, P.J., Xiang, B., Kim, H-M, Cao, J., Ha, K-J. 2018. Toward Predicting Changes in the Land Monsoon Rainfall a Decade in Advance J Climate, 31: 2699-2714. DOI: 10.1175/JCLI-D-17-0521.1

---

## Author Comment (AC3) · 9 Aug 2018

**Responses to Dr. Adams (Short Comments)**

I commend the authors, several of whom are my colleagues, efforts to attack this important problem, a problem that goes way beyond the scope of the North American Monsoon region. However, there are some fundamental problems with this paper that need to be addressed before publication. It is clear the study is fundamentally flawed by the conclusion that the Pacific Ocean off the coast of California is a moisture source for the monsoon. This is patently absurd given the water temperatures (10 to 20C) of the Pacific Ocean (see typical sounding Figure 2 below), no need to even consider the stable Subtropical Pacific High and nearly impassable mountains of California. So, just for this reason alone, all other results from this study should be called into question. However, the indication that the Sonoran and Mojave Desert could be a major moisture source through also stretches the limits of credibility– one of the driest, warmest regions on Earth that receives scares, very localized rainfall during the Monsoon becoming an important local source. These results are surely a result of the FLEXPART modeling employed which is apparently inadequate for the task at hand.

We thank Dr. Adams for his valuable time. Please find the response to your comments:

The Pacific Ocean off the coast of California together with the Gulf of California (GoC) is a moisture source region for the NAM. We have joined the two regions for simplicity, although the contributions from the latter region are higher. It is well known that the moisture transport from the GoC can be an important source for the North American Monsoon (NAM) precipitation. Please, find the references in our introduction section.

Previous works also highlight the importance of water vapour of NAM region itself in which the Sonoran desert is mostly included. Vegetation greening starts around two weeks after the rainfall and causes evapotranspiration to increase. Pleased find some references in our introduction. In fact, it is noteworthy that one of the reviewers of this manuscript (Dr. Vivoni) has recommended more insightful literature about this topic (Vivoni et al, 2008; Mendez-Barroso et al., 2010; Xiang et al., 2018).

The Mojave Desert in the southwest U.S. is a region of mean positive divergence of water vapour during summer at climatological scale, and constitutes a moisture source. To explain divergence of water vapour over land prevailing under a large subtropical anticyclone is difficult. In this case, surface and underground flows from less arid regions must supply the water required to counterbalance the observed excess of evaporation over precipitation (Peixoto and Oort, 1992).

These radical, game-changing conclusions that the authors arrive at, contradicting nearly 4 decades of the vast majority of studies (both observational and modeling), need to provide extraordinary evidence (particularly direct observational support, not modeled products of water vapor or precipitation products over Mexico) for their conclusions. Below I outline basic points that should be addressed in order to validate their conclusions.

Our introduction condenses a vast amount of previous studies on the NAM at **climatological scale** where it becomes clear that the conclusions of this study are no 'game-changing conclusions'.

(1) From simple thermodynamic arguments (the water vapor scale height is only about 2km), mid-level cannot be responsible for for the in the NAM region west of the elevated terrain (SMO, Continental Divide, etc). This is shown by Adams and Souza (2009), Maddox et al. 1995, Mazon et al. 2016, Rogers et al. 2017 and many, many others. Low-level water vapor (below 800mb) is required for generating convective instability over the valleys and low deserts. Elevated water vapor sources (above ~ 700mb) can help with initiation over elevated regions (e.g. SMO or Four-Corners region) but this convection cannot migrate towards lower regions without the support of low-level moisture. Entrainment of dry air is too detrimental. Gulf of Mexico can certainly contribute to NAM convective precipitation, but can not "cause" or be "responsible" for a sizeable portion of it, west of the SMO/Continental Divide regions. No low-level moisture, no deep convective precipitation in low-lying zones. Likewise, no strong surface heating, no deep convection (higher level moisture can actually be detrimental to the later, soundings can be too moist (see Adams and Souza 2009)).

We don´t suggest that water vapor above 800 mb generates convective instability over NAM region west of the SMO. We don´t affirm that moisture transport form the GOM "cause" or is directly "responsible" of the convective precipitation over the NAM.

We state that the water vapor from the Gulf of Mexico (GOM), Caribbean Sea and the terrestrial region over northeastern Mexico can play an important role days before of the synoptic-scale heavy precipitation events during the monsoon peak. In fact, we accept that low-level water vapor transport from the GOC can supply moisture at day 0 to provide convective instability, as Schiffer and Nesbitt (2012) proposed (lines 25-27).

In summary, moisture from large-scale patterns could influence (1) pre-surge air masses and thus surge generation (what is important to address surge predictability), or (2) precipitation production during surges.
We will revise the manuscript to be sure that there isn't any possible misunderstanding.

(2) The Model. The authors will have to provide more evidence as to how their extremely simplistic model (FLEXPART) which can not account for vertical transport inconvective updrafts, (scale 100s of meters to kms on the high end of organized convection, and order minutes to an hour), nor for entrainment/mixing in a realistic manner, and no need to mention the proper formation of hydrometeors and precipitation processes. Even assuming FLEXPART is a valid model for doing the impossible, consider also the arguments of Ana Maria Quesada. The evaporation/precipitation relationship is entirely space/time resolution dependent (See our figure 3 below, local surface fluxes have little relationship with total column, precipitable water vapor).

It is important to emphasize that we are working from a climatological perspective. We are providing a climatology of the long-range water vapor transport aimed toward the NAM.

Since the last decade, an increasing number of studies have taken advantage of the use of FLEXPART to analyze the water transport towards different regions. The accuracy and robustness of this approach have facilitated the assessment of the averaged values of moisture sources in various climatic regions as the Iberian Peninsula (Gimeno et al., 2010) or the Danube River Basin (Stojanovic et al., 2017; Ciric et al., 2017). But FLEXPART model have also provided insightful information about the moisture transport toward tropical regions as Centro America (Durán-Quesada et al., 2010; Durán-Quesada et al., 2017), Colombia (Hoyos et al., 2018), Sahel (Nieto et al., 2006), India (Ordonez et al., 2012) or the Amazon region (Drumond et al., 2014; Sorí et al., 2018) where convective precipitation dominates.

We are not studying local processes related to the evaporation/precipitation relationship. Regarding the arguments of the reviewer #2 Dra. Duran-Quesada, please find our response to her comments.

(3) Data. ERA-Interim data is not adequate. Reanalysis data, in general, is inadequate because model-generated values depend on convective and microphysical parameterizations. Pressure, geopotential heights and winds, can be dynamically constrained, the water vapor distribution cannot. This poorly measured quantity is extremely difficult to replicate in the NAM region (see Radhakrishna et al. 2016 and many others) (Also, See attached figure 3 for ERA-Interim vs GPS Precipitable water vapor data below). ERA-5 is still bad, but much better than ERA-Interim or NARR. Given the radically different nature of surface evaporation of the oceanic (e.g., different surface wind speeds and temperatures) zones and more critically over the complex topography of Mexico and the Southwest U.S., how is the evaporation determined at the necessary spatial/temporal resolution.

In this work, the atmospheric water arriving to the NAM region at climatological scale is studied by using the Lagrangian particle model FLEXPART. Therefore:

(1) A gridded product to run FLEXPART is necessary. ERA-Interim reanalysis is used as this dataset has been found to provide a reliable representation of the atmospheric branch of the hydrological cycle (Lorenz and Kunstmann 2012; Trenberth et al. 2011).

(2) Typically, in climatology a 30 years period is considered as an appropriate period of time to average variations in weather and evaluate climatic effects. GPS precipitable water vapor (PWV) data over the Mexican NAM region starts around 2012. It is then obvious that the available temporal span is still too short for estimating a reasonably proper climatology.

PWV values observed in the GPS sites could reflect diurnal variations related to mesoscale processes or the complex terrain, which obviously could not be analysed in a 1ºx1º daily reanalysis product. However, this is not the topic of our work.

(4) Moisture recycling. Just a point of logic, if local moisture recycling is important then (1) how does the monsoon precipitation begin? Vegetation green up occurs at the end of July into August (2) Why does precipitation decrease, become for more variable around the second week of August (see for example Kursinski et al. 2008) in Northwestern Mexico and Arizona even though vegetation is green and surface moister, in general? Also, wet surfaces can actually be detrimental to deep convection, as convective temperatures cannot be reached for this region (see Kirsten Findell's 2011 article among others). Our results from the GPS Hydromet Network show that at the sub- diurnal scale, local water vapor fluxes and precipitable water vapor have small correlations, lagged or not. So at least locally, it is not apparent that moisture recycling (see figure 3 below) plays any important role in column water vapor. Boundary layers are extremely deep and well-mixed over much of the region, so very local surface evaporation would be mixed out quickly, not providing sufficient "density" to either contribute to increasing convective instability, nor precipitation efficiency. Only large-scale advection could account for generating the necessary water vapor fluxes to produce the instability necessary to generate convective activity.

(E-P)>0 values over the NAM itself have been roughly approximated to the recycling ratio but it is not strictly correct. We will delete the term "recycling" overall the manuscript and instead of that, we will use "inland evaporative sources".
Please note that this work is not about subdiurnal and local scales but climatological and monsoonal scales.

(5) And probably most important and what makes this a very difficult problem to separate sources easily is the mixing over the SMO due to deep convective activity. Moist air to the east of Arizona and the SMO, in general, may result from deep convective mixing further south and then transported northward along the high terrain with or without precipitation.

We don't find an allusion to our study in this comment.

**Minor Comments**

"Later studies claimed" is not the correct word, better "indicated" or something of the like. The vast majority of studies over the last several decades have shown that the time mean as well as transient flow are dominated by the EPac and GoC.

The context of "later studies" is after the 50s and 60s decades.

"While compelling, these results are based on observations from a limited field campaign". These are a least real-world data and not model dependent as are ERA-Interim.

Fortunately, for large-scale climatologists, there exist global PWV products as NVAPM (Vonder Haar et al., 2012) that incorporates observations from a variety of surface and space borne sensors including GPS, supporting a more reliable climate analysis.

ERA-Interim dataset has given a reasonable estimate of PWV when compared against GPS data at daily, monthly, and seasonal scales in several regions as Gana (Acheampong et al., 2015), United States (Bordi et al., 2016) or the Tibetan Plateau (Wang et al., 2017) among others. The agreement is better at monthly (figure R1), and seasonal scales, but in general, the intraseasonal variability, which is important for the present work, is captured properly.

[Figure]

**Figure R1.** Comparison between monthly PWV measured by Arizona GPS station (red line) and that by ERAI reanalysis (blue line). Source: Fig 6 from Bordi et al. (2016).

"IVs, as well as improved models of the flow over complex terrain like that of the NAM region are both needed to better understand the role of IVs in supporting convective outbreaks across the monsoon region." This is sort of a throwaway statement. Our understanding of IV is fairly good and there dynamic effects (increased shear) appears to be responsible for convective organization. Water vapor transport plays no role, nor its advection, as these features are found at 300 or 200mb. (See Finch and Johnson2010).

Finch and Johnson (2010) analyzed one significant IV that passed over northwestern Mexico from 10 to 13 July 2004. One case of study, in which limited data from NAME were examined (Newman and Johnson, 2012).

We find surprising that Dr. Adams declares to understand the processes related to the role of IVs in supporting convective outbreaks being coauthor of Lahmers et al. (2016) where the following statement can been literally found: "the exact roles of synoptic and mesoscale processes associated with IVs and convective organization are not entirely understood".
Maybe more exploration is needed for a better understanding.

IVs seem to help the moist air to bypass the Sierra Madre westward accompanied by organized convection and upward vertical velocity over the NAM region. You need to back this claim up. As shown by Finch and Johnson (2010). windshear in organizing convection is important, adiabatic lifting is weak. From our examination of Marty Ralph and Tom Galarneau's results (using lightning data as a proxy) Invited Manuscript for Atmospheric Research there is little evidence of low-level water vapor transport from the Gulf of Mexico on strong convective days. Strong shear can certainly be important in the easterly winds, but we should not convolve dynamics and water vapor transport. "This wind reversal is not of sufficient magnitude and scale to meet the criteria of Ramage (1971) for a monsoon (Hoell et al. 2016)." You need more the two citations to make the claim that it is not a monsoon circulation. "Bosilovich et al. (2003) found the dominant sources of monsoon precipitation to be the local evaporation and transport from the tropical Atlantic Ocean (including the GOMand Caribbean Sea)." As Bosilovich et al. (2003) note in their study "It is also worth while to reiterate that the model does not resolve the Gulf of California, which should influence the sources of water". Always critical to consider space/time resolution of large-scale models.

Finch and Johnson (2010) analysed only one case of study….

Regarding the affirmation, "*This wind reversal is not of sufficient magnitude and scale to meet the criteria of Ramage (1971) for a monsoon (Hoell et al. 2016)*."

This is simple. Ramage (1971) defined a global monsoon domain based on the seasonal reversal of large-scale lower tropospheric wind. Such monsoon domain is found over the Eastern Hemisphere as shows the figure R2:

[Figure]

**Figure R2.** Domain of monsoons of Ramage (1971). Source: Krishnamurti et al. (2013).

The seasonal wind reversal along the Pacific coast of the NAM do not satisfy this global definition because occurs below the boundary layer. That is to say, the scale of the onshore circulation is not enough to meet the criteria of Ramage (1971).

Moreover, the monsoonal behaviour of this region is not being discussed in this work, but its name. Please see the comment # 1 of the reviewer Dra. Duran-Quesada.

Our work with Chris Castro and others demonstrates the need for Convective Resolving Models in order to capture mesoscale convective systems (order kms resolution), which are responsible for a large portion of precipitation particularly in NW Mexico. Results from low resolution models 1degree x 1 degree should be critically assessed given their inability to produce these systems correctly. (see Lamhers et al. 2016,Luong et al. 2017, Moker et al in press JAMC)

Taking into account the objective of this study, to resolve mesoscale convective systems is not necessary. In this work, the origin of the atmospheric water arriving to the NAM during a 34-yr period is investigated by using a Lagrangian diagnosis method. Rainfall events over the NAM are identified at synoptic scale, being representative of a large area of the studied region. We explore if the identified moisture sources play a (direct or indirect) role on such scale rainfall development.

**References**

Acheampong, A.A., Fosu, C., Amekudzi, L.K., Kass, E. 2015. Comparison of precipitable water over Ghana using GPS signals and reanalysis products. *J. Geod. Sci.*; 5: 163–170. DOI 10.1515/jogs-2015-0016.

Bordi, I., Xiuhua, Z., Fraedrich, K. 2016. Precipitable water vapor and its relationship with the Standardized Precipitation Index: ground-based GPS measurements and reanalysis data. *Theor Appl Climatol.*, 123: 263-275. DOI 10.1007/s00704-014-1355-0

Ciric, D., Nieto, R., Ramos, A.M., Drumond, A., Gimeno L. 2017. Wet Spells and Associated Moisture Sources Anomalies across Danube River Basin, *Water*, 9, 615, doi: 10.3390/w9080615. Water 2018, 10(6), 738;

Drumond, A., Marengo, J., Ambrizi, T., Nieto, R., Moreira, L., Gimeno L. 2014. The role of Amazon Basin moisture on the atmospheric branch of the hydrological cycle: a Lagrangian analysis, Hydrology and Earth System Sciences, vol. 18, pages 2577-2598, doi: 10.5194/hessd-18-2577-2598

Durán-Quesada, A.M., Gimeno, L., Amador, J.A., Nieto R. 2010. Moisture sources for Central America: Identification of moisture sources using a Lagrangian analysis technique, *J. Geophys. Res.*, 115, D05103, doi: 10.1029/2009JD012455

Durán-Quesada, A.M., Gimeno, L., Amador, A. 2017. Role of moisture transport for Central American precipitation. *Earth Syst. Dynam.*, 8, 147–161.

Finch, Z.O. and Johnson R.H. 2010. Observational Analysis of an Upper-Level Inverted Trough during the 2004 North American Monsoon Experiment. *Mon. Wea. Rev.*, 138, 3540 – 3555.

Gimeno, L., R. Nieto, R. M. Trigo, S. Vicente-Serrano, and J. I. López- Moreno (2010), Where does the Iberian Peninsula moisture come from? An answer based on a Lagrangian approach, *J. Hydrometeorol.*, 11, 421– 436, doi:10.1175/2009JHM1182.1.

Hoyos, I., Domínguez, F., Canon-Barriga, J., Martínez, J.A., Nieto, R., Gimeno, L., Dirmeyer, P.A. 2018. Moisture origin and transport processes in Colombia, northern South America. *Clim Dyn.*, 50: 971–990.

Krishnamurti, T.N., Stefanova, L., Misra, V. 2013.Tropical Meteorology. An Introduction. ISSN 2194-5225 (electronic). ISBN 978-1-4614-7408-1 ISBN 978-1-4614-7409-8 (eBook). DOI 10.1007/978-1-4614-7409-8.
Springer Science+Business Media New York

Lahmers, T., Castro, C.L., Adams, D.K., Serra, Y.L., Brost, J.J., Luong, T. Long-Term Changes in the Climatology of Transient Inverted Troughs over the North American Monsoon Region and Their Effects on Precipitation. J. Climate, 29: 6037 – 6064. DOI: 10.1175/JCLI-D-15-0726.1

Lorenz, C., Kunstmann, H. 2012. The hydrological cycle in three state of- the-art reanalyses: Intercomparison and performance analysis. J Hydrometeor 13 (5): 1397-1420. doi:10.1175/jhm-d-11-088.1

Mendez-Barroso, L.A., and Vivoni, E.R. 2010. Observed Shifts in Land Surface Conditions during the North American Monsoon: Implications for a Vegetation-Rainfall Feedback Mechanism. Journal of Arid Environments. 74(5): 549-555.

Newman, A., and Johnson, R.H. 2012. Mechanisms for Precipitation Enhancement in a North American Monsoon Upper-Tropospheric Trough. *J. Atmos. Sci.*, 69, 1775–1792,doi:10.1175/JAS-D-11-0223.1.

Nieto, R., L. Gimeno, and R. M. Trigo (2006), A Lagrangian identification of major sources of Sahel moisture, Geophys. Res. Lett., 33, L18707, doi: 10.1029/2006GL027232.

Ordoñez, P., Ribera, P., Gallego, D., and Peña-Ortiz. C.: Major moisture sources for Western and 5 Southern India and their role on synoptic-scale rainfall events, Hydrol. Process., 26 (25), 3886 - 3895, doi: 10.1002/hyp.8455, 2012.

Peixoto, J.P., and Oort, A.H.: Physics of climate. Springer, Berlin, 1992.

Sorí R., Marengo, J.A., The Atmospheric Branch of the Hydrological Cycle over the Negro and Madeira River Basins in the Amazon Region. *Water* 2018, *10*(6), 738; https://doi.org/10.3390/w10060738.

Trenberth, K.E., Fasullo, J.T., Mackaro, J. 2011. Atmospheric moisture transports from ocean to land and global energy flows in reanalyses. *J Clim* 24 (18): 4907–4924. doi:10.1175/2011jcli4171.1

Schiffer N.J. and Nesbitt S.W. 2012. Flow, Moisture, and Thermodynamic Variability Associated with Gulf of California Surges within the North American Monsoon. *J. Climate*, 25, 4220- 4241. doi: 10.1175/JCL I-D-11-00266.1.

Stojanovic, M., Drumond, A., Nieto, R., Gimeno, L. 2017. Moisture Transport Anomalies over the Danube River Basin during Two Drought Events: A Lagrangian Analysis, *Atmosphere*, 8(10), 193; doi: 10.3390/atmos8100193.

Vivoni, E.R., Moreno, H.A., Mascaro, G., Rodriguez, J.C., Watts, C.J., Garatuza-Payan, J., and Scott, R.L. 2008. Observed Relation between Evapotranspiration and Soil Moisture in the North American Monsoon Region Geophys. Res. Lett., 35: L22403, dos: 10.1029/2008GL036001.

Vonder Haar, T.M., Bytheway, J.L., and Forsythe, J.M. 2012. Weather and climate analyses using improved global water vapor observations. *Geophysical Research Letters*, 39, 1–6, L15802. doi:10.1029/2012GL052094

Wang, Y., Yang, K., Pan Z., Qin, J., Chen, D., Lin, C., Chen, Y., Lazhu, Tang, W., Han, M., Lu, N., Wu, H. 2017. Evaluation of Precipitable Water Vapor from Four Satellite Products and Four Reanalysis Datasets against GPS Measurements on .the Southern Tibetan Plateau. J Climate, 30: 5699- 5713.

Xiang, T., Vivoni, E.R., and Gochis, D.J. 2018. Influence of Initial Soil Moisture and Vegetation Conditions on Monsoon Precipitation Events in Northwest Mexico. Atmosfera. 31(1): 25-45.

---

## Author Response (AR1)

[revised manuscript text omitted]

---

## Author Response (AR2)

Dear Editor,

Please, find a point-by-point response to the reviews and a marked-up manuscript version.

**Minor comments anonymous Referee #3**

Page 2, Lines 15-16: The sentence is really confuse, please rewrite it.
Page 2, Line 18: ... The onset of NAM...
Lines 15 - 16 and line 18 have been rewritten. Now lines 14-21.
Page 3, L6: Definition of GOC.
GOC was defined in Page 2, L21. Now page 2, L18.
Page 16, L 19: Which Journal was this paper published?
JGR-A, it is included in page 16, L18.

**Minor comments anonymous Referee #4**

1- Add 'backward' in line 31 of p 3
Added, now line 29 of p3
2- Delete 'it' in line 30 od p 4
Done
3- Figure 2 does not show the monthly precipitation distribution. It shows the monthly distribution of precipitation events, which is not the same thing
We have changed it. See line 30, p6.
4- I think that the color scale in fig 3 is not appropriate, it uses blue colors for the dry areas and redish for the wettest ones. I think that is misleading and the color scale should be inverted.
Done
5- Add 'the' after during in line 13 of p 7
Added. Now line 10, p7.
6- Arrows are difficult to see in figure 5
We have improved the figure.
7- The monsoon onset usually occurs in July, so rephrase the first sentence in line 8 of p 9.
Done. Line 1, p9.
8- Holdover should be hold over in line 18 of p 9
Done. Line 11, p9
9- Line 24, the figure with the E-P values is figure 4, not 5, as incorrectly written
It was figure 7. Changed in line 18.
10- Lines 1 to 23 need rephrasing, since the explanation that they provide about figure 8 is confusing, contains some grammar errors and no clear message comes from them.
Done. See line 27 (p9) to line 4 (p10).
11- Figure 3 should be figure 10 in line 32 of p 10
It was figure 3, but this sentence wasn't necessary, we have deleted for clarity.
12- Lines 1-12 in p 11 need to be rephrased and clarified
Line 26 (p10) - line 4 (p11)
13- I would need an explanation on how the authors see the transient lows (easterly waves or tropical cyclones) off the west Mexican coast in figure 11. The comment of this figure should be more careful

Lines 9 - 10, p11.
14- Add 'that' after suggests in line 1 p12
Added. Line 29, p11.
15- Add 'climatological' before 'large-scale' in line 7 p 12

5  Added. Line 4, p12
16- I think that the term 'most Eastern part of the Northern Pacific' is misleading, it refers just to one of the identified areas within the study region, but not to the actual N Pacific.
Changed by "an oceanic adjacent area". Line 9, p12.
17- Change recycling in line 11 of p 13

10  Changed by "the region itself". Line 7, p 13.

15  .

[revised manuscript text omitted]

[Figure]

**Figure 6. Name and geographic limits of the moisture sources defined for the WNAM region.**

[Figure]

5        **Figure 7. Monthly $(E - P)_{1-6}$ percentages for the six areas defined as moisture sources.**

[Figure]

10 **Figure 8. JAS time series of $(E - P)_n$ (n=1 to 6) integrated over (a) WNAM, (b) NE-MEX, (c) GOM-CAR, (d) SW-US, (e) NEP and (f) GOC. Solid line: wet days. Dotted line: dry days. Note that the panel a is scaled by 0.5.**

[Figure]

**Figure 9. Anomalies of (E − P)₁, (E − P)₂, and (E − P).₅ during JA (1981–2014) for extreme rainfall days minus low rainfall days. Unit: mm/day.Black line delineates the study region.**

[Figure]

**Figure 10. JA time series of (E-P)$_n$ (n=1 to 6) integrated over (a) WNAM, (b) NE-MEX, (c) GOM-CAR and (d) GOC. Black solid line: extreme, dashed line: moderate and dotted line: weak rainfall events. Note that the panel a is scaled by 0.5.**

[Figure]

**Figure 11. Composites at of geopotential height anomalies (colors, m) and moisture transport anomalies (arrows, kg kg$^{-1}$m s$^{-1}$)with respect to regional-scale dry events over the WNAM during the monsoon peak (July and August) for (a) weak, (b) moderate and (c) extreme precipitation events at 200 hPa and (d) weak, (e) moderate and (f) extreme precipitation events at 700 hPa.**